# Selective compression learning of latent representations for variable-rate image compression

**Jooyoung Lee**[1,2], **Seyoon Jeong**[2], **Munchurl Kim**[1]*

[1]School of Electrical Engineering, Korea Advanced Institute of Science and Technology, Korea
[2]Electronics and Telecommunications Research Institute, Korea
umpu@kaist.ac.kr, jsy@etri.re.kr, mkimee@kaist.ac.kr

## Abstract

Recently, many neural network-based image compression methods have shown promising results superior to the existing tool-based conventional codecs. However, most of them are often trained as separate models for different target bit rates, thus increasing the model complexity. Therefore, several studies have been conducted for learned compression that supports variable rates with single models, but they require additional network modules, layers, or inputs that often lead to complexity overhead, or do not provide sufficient coding efficiency. In this paper, we *firstly* propose a selective compression method that partially encodes the latent representations in a fully generalized manner for deep learning-based variable-rate image compression. The proposed method adaptively determines essential representation elements for compression of different target quality levels. For this, we first generate a 3D importance map as the nature of input content to represent the underlying importance of the representation elements. The 3D importance map is then adjusted for different target quality levels using importance adjustment curves. The adjusted 3D importance map is finally converted into a 3D binary mask to determine the essential representation elements for compression. The proposed method can be easily integrated with the existing compression models with a negligible amount of overhead increase. Our method can also enable continuously variable-rate compression via simple interpolation of the importance adjustment curves among different quality levels. The extensive experimental results show that the proposed method can achieve comparable compression efficiency as those of the separately trained reference compression models and can reduce decoding time owing to the selective compression. The sample codes are publicly available at https://github.com/JooyoungLeeETRI/SCR.

## 1   Introduction

Recently, neural network (NN)-based image compression methods [1, 2, 3, 4, 5, 6, 7, 8, 9, 10, 11, 12, 13] have been actively studied and shown superior performance in terms of PSNR BD-rate to those of the conventional tool-based compression methods, such as BPG [14] and JPEG2000 [15]. A few recent methods [11, 12] achieved comparable results with respect to the state-of-the-arts codec, called H.266 Intra-coding [16]. However, since most of the previous deep learning-based models are trained separately for different target compression levels, several models with a large number of parameters are required to support various compression levels. To address this issue, recently several methods [1, 17, 18, 19, 20, 21, 22], which use conditional transform or adaptive quantization, have been proposed. However, most of them require additional network modules, layers, or inputs that may cause complexity overhead. In this paper, a novel 'selective compression of representations' (SCR) method is presented, which performs entropy coding only for the partially selected latent

---

*Corresponding author

36th Conference on Neural Information Processing Systems (NeurIPS 2022).

representations. The selection of representations is determined via a 3D binary mask generation process in a target quality-adaptive manner. In the 3D binary mask generation of our SCR method (see Figure 1b), (i) a 3D importance map of the same size, *independent of target quality levels*, is generated for the 3D latent representations (multi-channel feature maps); (ii) the 3D importance map is adjusted via the channel-wise importance adjustment curves for a given target quality level; and (iii) the 3D binary mask is then generated by taking the round-off the adjusted 3D importance map. Note that the *target-quality-independent* 3D importance map becomes *target-quality-dependent* after the channel-wise importance adjustment. We incorporate our method with an adaptive quantization scheme [19] into several existing deep learning-based reference compression models [6, 7], where we jointly optimize the whole elements together with our selective compression of latent representations and adaptive quantization in an end-to-end manner. In the architectural aspect, our SCR method minimizes the overhead by utilizing only a single $1 \times 1$ convolutional layer to generate the 3D importance map and importance adjustment curves for a finite number of target quality levels. In addition, the SCR method also supports continuously variable-rate compression through a simple non-linear interpolation of the importance adjustment curves between two discrete target quality levels. Furthermore, the SCR method significantly reduces the decoding time compared to the existing lightweight variable-rate methods [19, 20, 23] by skipping the entropy decoding process for a substantial amount of unselected representations. Impressively, the coding efficiency of the proposed SCR method is better or comparable to those of the separately trained reference compression models [6, 7], for different target quality levels, and is superior to those of the existing variable-rate compression methods [19, 20, 23, 24, 25]. The contributions of this study are summarized as follows:

- To our best knowledge, the proposed SCR method is the first NN-based variable-rate image compression method that selectively compresses the representations in a fully generalized way and a target quality-adaptive manner. It provides compression efficiency comparable to those of the separately trained reference compression models.

- The proposed SCR method can be applied to various existing image compression models without modifying their architectures, thus allowing for a high applicability. We incorporate very lightweight modules for SCR, including only a single $1 \times 1$ convolutional layer and a small number of importance-adjustment curves, into the existing compression models. Our SCR method even reduces a decoding time compared to those of the existing lightweight variable-rate models and high bit rate reference compression models owing to the selective compression.

- To enable continuously variable-rate compression, the proposed SCR method extends the existing interpolation-based approach by additionally incorporating the interpolation of the importance adjustment curves between discrete quality levels in which the SCR model is trained. With experiments, we verify that our extension can stably support the continuous bit rate compression.

## 2 Related work

Several studies [1, 17, 18, 19, 20, 21, 22, 23, 26, 27] have been conducted for enabling a single NN-based image compression model to support variable-rate compression. The first learning-based variable-rate image compression model [1] progressively performs image compression in a low-to-high quality manner by accumulating additional binary representations as the number of compression iterations increases. For the entropy model-based approaches, Choi *et al.* [17] proposed a conditional convolution that adaptively operates according to different target quality levels. Scale and shift factors are derived from a one-hot vector representing a quality level, and are then applied to each output of convolutional layers. Cai *et al.* [18] stacks representations to form a multiscale structure, and then determine point-wise scales, representing the importance levels of representations to be compressed. This method also uses additional modules named a multi-scale decomposition transform layer (MSD) and an inverse multi-scale decomposition transform layer (IMSD), both of which consist of multiple convolutional layers. Cui *et al.* [19] utilizes two types of vectors for the adaptive quantization and inverse quantization, and Chen *et al.* [23] similarly utilizes scaling and shifting vectors. These methods [19, 23] support variable-rate compression with negligible overhead on top of the existing compression models ([6, 7] for Cui *et al.* [19] and [3, 6, 10] for Chen *et al.* [23]). However, as described in our experimental results and Rippel's work [20], the adaptive quantization alone does not provide sufficient coding efficiency compared to those of the separately trained models. Lu *et al.* [21] use an adaptive quantization layer (AQL) and an inverse adaptive quantization layer (IAQL) to obtain quantization and inverse quantization factors for each representation. Here, the AQL and

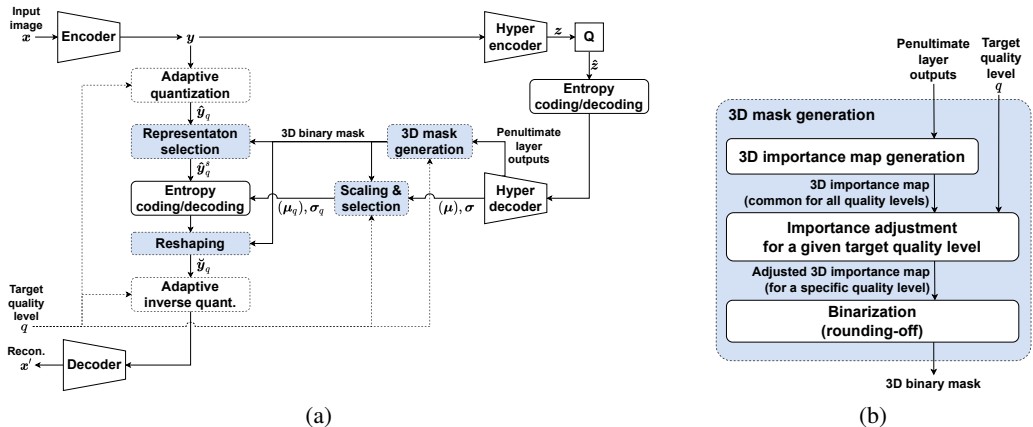

Figure 1: (a) Overall architecture of the proposed SCR method. In this figure, the SCR method is incorporated into the Hyperprior [6] model. The elements for the variable-rate compression are represented as the dotted boxes, and especially those for the selective compression are highlighted in light blue. (b) The 3D binary mask generation process of the SCR method.

IAQL layers are separately trained for each target quality level. Rippel *et al.* [20] additionally feed the information of a target quality level into each convolutional layer in the form of a "level map" where a one-hot vector of 8 quality levels is assigned for all spatial positions. The input level map can make each convolutional layer adaptively work according to a given target quality level. Lu *et al.* [26] firstly presented an NN-based quality scalable coding scheme, named PLONQ, using nested quantization and latent ordering. However, it could not achieve comparable results to its separately trained base compression model [7]. Song *et al.* [22] supports the image compression of spatially different qualities using a spatial quality map. Although this method provides a new functionality, the additional module, named spatially-adaptive feature transform, that transforms the spatial quality map into an input for each convolutional layer may increase the overall complexity.

From the perspective of partial encoding for representations, Li *et al.* [24] and Mentzer *et al.* [25] utilize 2D importance maps to represent spatial importance of representations, which allows for spatially different bit allocation in different regions. Their models mainly focus on the fixed-rate compression, although Mentzer *et al.* [25] presented a few decoded images at multiple-rates using the shared en/decoder networks. It should be also noted that their 2D importance maps convey the information about how many representation elements at each spatial location should be taken forward along the channel (See Figure 15 in Appendix F). On the other hand, our component-wise 3D importance map represents the essence of individual representations, which is adaptively adjusted according to given target qualities in an R-D optimization sense. This brings a good generalization with higher-coding efficiency and more stability in training, as further discussed in Appendix F.

## 3 Proposed method

### 3.1 Overall architecture

Our SCR method can be combined with adaptive quantization such as Cui *et al.* [19] and Chen *et al.* [23], on top of several compression architectures with hyper-encoder and hyper-decoder [6] such as the models in [6, 7, 8, 9, 10, 11, 12], as shown in Figure 1a. In this paper, we apply our SCR method for several reference compression models, Hyperprior [6], Mean-scale [7], and Context [7], to show its effectiveness with generality. In the architecture with hyper-encoder and hyper-decoder [6], input image $x$ is transformed into a representation $y$ using an encoder network, and the hyper encoder and decoder are used to code the distribution parameters for the quantized representation $\hat{y}$ of $y$ as a side information, with which $\hat{y}$ is entropy-coded and decoded. The representation $\hat{y}$ is then reconstructed into an image $x'$ through a decoder network. Upon this base compression architecture, we exploit two additional elements: adaptive quantization and selective compression to enable variable-rate compression. The selection of representation elements in the encoder side is expressed as follows:

$$\hat{\boldsymbol{y}}_q^s = M(\hat{\boldsymbol{y}}_q, m(\hat{\boldsymbol{z}}, q)), \text{ with } \hat{\boldsymbol{y}}_q = AdaQ_q(\boldsymbol{y}), \tag{1}$$

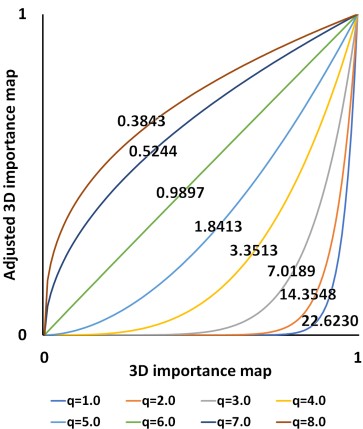

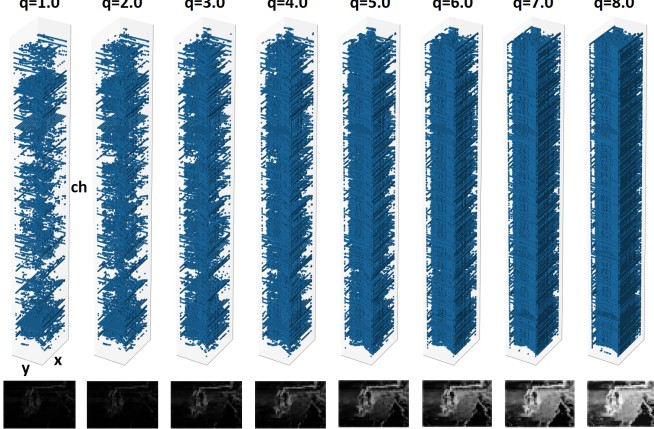

Figure 2: Average importance adjustment curves for each target quality level $q$ in our SCR model (on Hyperprior [6]). The numbers indicate the mean values of trained $\boldsymbol{\gamma}_q$ vectors (See Section 3.2).

Figure 3: Top - sample masks for 8 target quality levels where the dark blue indicates the selected representation elements by 3D binary masks. Bottom - the masks averaged along the channel axis. The higher the target quality is, the more the representation elements are selected, especially in more complex regions.

where $\hat{\boldsymbol{y}}_q^s$ is a set of the selected elements of $\hat{\boldsymbol{y}}_q$ for a given target quality level $q$ and $AdaQ_q(\cdot)$ is a target quality-adaptive quantization operator where $AdaQ_q(\boldsymbol{y}) = Round(\boldsymbol{y}/\boldsymbol{QV}_q)$ with a quantization vector $\boldsymbol{QV}_q$. $M(\cdot)$ is an element selection operator for $\hat{\boldsymbol{y}}_q$, and $m(\hat{\boldsymbol{z}}, q)$ represents a generated 3D binary mask for $q$ and quantized hyperprior $\hat{\boldsymbol{z}}$ (Sec. 3.2). The representation $\boldsymbol{y}$ is the output $En(\boldsymbol{x})$ of the encoder network $En(\cdot)$ for an input image $\boldsymbol{x}$ in Figure 1a. It should be noted that $\hat{\boldsymbol{y}}_q^s$ is entropy-coded and entropy-decoded using an entropy model based on a target quality dependent distribution $P_q$ (Sec. 3.3). The image reconstruction $\boldsymbol{x}_q'$ in the decoder side is given as:

$$\boldsymbol{x}_q' = De(AdaIQ_q(\breve{\boldsymbol{y}}_q)), \tag{2}$$
$$\text{with } AdaIQ_q(\breve{\boldsymbol{y}}_q) = \breve{\boldsymbol{y}}_q \cdot \boldsymbol{IQV}_q \text{ and } \breve{\boldsymbol{y}}_q = Re(\hat{\boldsymbol{y}}_q^s, m(\hat{\boldsymbol{z}}, q)),$$

where $\boldsymbol{x}_q'$ is a reconstructed image for a given target quality level $q$, as the output of the decoder network $De(\cdot)$, $AdaIQ_q(\cdot)$ is an adaptive inverse-quantization operator that multiplies an inverse-quantization vector $\boldsymbol{IQV}_q$ to input $\breve{\boldsymbol{y}}_q$, and $Re(\cdot)$ is a reshaping operator that converts the selected elements $\hat{\boldsymbol{y}}_q^s$ in a 1D shape into the elements of a 3D-shaped representation in place by using the 3D binary mask $m(\hat{\boldsymbol{z}}, q)$. For the unselected elements, the reshaping operator $Re(\cdot)$ places 0 values in the corresponding positions. It should be noted that the unselected elements are filled with 0 values for the Mean-scale [7] and Context [7] models that utilize $\boldsymbol{\mu}$ estimation as well as for the Hyperprior [6] model. Example source codes for $M(\cdot)$ and $Re(\cdot)$ are provided in Appendix A. In Eqs. 1 and 2, the vector dimensionalities of $\boldsymbol{QV}_q$ and $\boldsymbol{IQV}_q$ are both $C_{\boldsymbol{y}}$, the number of channels in $\boldsymbol{y}$, so that the quantization of $\boldsymbol{y}$ and the inverse quantization of $\breve{\boldsymbol{y}}_q$ are performed channel-wise by the respective elements of $\boldsymbol{QV}_q$ and $\boldsymbol{IQV}_q$, respectively, as in the previous adaptive quantization method [19].

### 3.2 3D binary mask generation

The 3D binary mask generation process consists of the three steps: (i) 3D importance map generation, (ii) importance adjustment, and (iii) binarization, as shown in Figure 1b, which is given as:

$$m(\hat{\boldsymbol{z}}, q) = B(im(\hat{\boldsymbol{z}})^{\gamma_q}) \tag{3}$$

where $im(\hat{\boldsymbol{z}})$ is a 3D importance map that is generated via the hyper-decoder for the hyperprior $\hat{\boldsymbol{z}}$ as input, $\boldsymbol{\gamma}_q = \left[\gamma_q^1, \gamma_q^2, ..., \gamma_q^N\right]$ is a parameter vector of dimensionality $N(= C_{\boldsymbol{y}})$ where the parameters are learned to determine the channel-wise importance adjustment curves for a given target quality $q$, and $B(\cdot)$ is a binarization operator with the rounding-off.

**3D importance map generation**. The 3D importance map $im(\hat{\boldsymbol{z}})$, which has values in the range between 0 and 1, represents the underlying importance of each element in $\boldsymbol{y}$. Note that the 3D importance map is generated, not dependently of target quality levels but dependently of input images, thus it represents the nature of $\boldsymbol{y}$ in perspective of element-wise importance. Without utilizing a dedicated complex network that generates $im(\hat{\boldsymbol{z}})$, we feed the outputs of the penultimate

convolutional layer (after the activation) in the hyper decoder into a single $1 \times 1$ convolutional layer in the mask generation module, followed by a clipping function to obtain the importance values between 0 and 1. To be specific, along with the the single $1 \times 1$ convolutional layer above, the hyper en/decoder networks also play a role of compressing and reconstructing (generating) the 3D importance map. However, even for the methods without hyper en/decoder networks, our method can also be applied if a dedicated network to the compression and generation of 3D importance map is adopted. In this case, we expect that the inherent compression efficiency due to the selective compression can be maintained because the bitstream for the auxiliary information is relatively very small. Since the compressed hyperprior bitstreams in the current SCR model often take about $2 \sim 3\%$ of the total bit rates, the auxiliary bitstreams only for the 3D importance map are expected to be less than this. Further study on the dedicated en/decoder structures may lead to additional performance improvement.

**Importance adjustment**. The actual importance of each representation element may vary according to various target quality levels. For example, some representation elements corresponding to the texture of high complexity in the images may not be necessarily required in low-quality compression. Thus, it is natural to adjust the 3D importance map, which is commonly used for all quality levels, according to a specific target quality level. For this, we devise a scheme of adjusting the 3D importance map $im(\hat{z})$ using importance adjustment curves for various target quality levels. The importance adjustment curves change the element values of $im(\hat{z})$ channel-wise where their curvatures are learned as a parameter vector $\boldsymbol{\gamma}_q$ for $1 < q < N_Q$ where $N_Q$ is a total number of target quality levels. Note that the target quality improves as $q$ increases. Figure 2 shows average importance adjustment curves for each target quality level $q$. In Figure 2, the horizontal and vertical axes represent the input $im(\hat{z})$ value to be adjusted and its adjusted result, respectively, and the numbers labeled on the importance adjustment curves indicate the average values of trained $\boldsymbol{\gamma}_q$ vectors for $N_Q$ target quality levels. It is noted in Figure 2 that the importance adjustment curves for $q > 6$ tend to amplify the elements of input $im(\hat{z})$ while attenuating them for $q < 6$ in an average sense. For $q = 6$, there is little variation with an average $\bar{\gamma}_q = 0.9897$ before and after the importance adjustment. Consequently, $im(\hat{z})$ is more strongly amplified in an overall sense for higher target quality levels. Whereas, for the lower target quality levels, $im(\hat{z})$ is largely attenuated in general, so only a small number of $im(\hat{z})$ elements whose values are close to 1 can maintain their importance. It should be noted in Figure 2 that, although the $\bar{\gamma}_q$ values are monotonic with the change in the target quality level $q$, the individual elements in the $\boldsymbol{\gamma}_q$ vectors are not always the cases because some representations are optimized in use only for a lower bit rate and can be ignored or de-emphasized in a higher bit rate range, as shown in Figure 5. The total number of $\boldsymbol{\gamma}_q$ vectors is $N_Q$, thus a total of $N_Q \times C_{\boldsymbol{y}}$ parameters are learned for all $\boldsymbol{\gamma}_q$ vectors. In our implementation, $N_Q$ is set to 8 and $C_{\boldsymbol{y}}$ is set to that of the original reference model.

**Binarization**. The 3D binary mask is finally determined by the rounding operator, denoted as $B(\cdot)$. Here, "1" values in the output 3D binary mask indicate that the corresponding elements in $\boldsymbol{y}$, at the same coordinates, are selected for compression. Figure 3 shows examples of the generated masks for different target quality levels from $q = 1.0$ to $q = 8.0$ when our SCR method is implemented on top of the Hyperprior [6] model and Kodim12 image of the Kodak image set [28] is used as an input sample, in which the components marked in dark blue indicate "1" values. When $q$ is set to 1.0, the lowest quality level in our SCR method, only $3.22\%$ of the total elements are selected, and the selection ratio gradually increases as the $q$ value increases. For $q = 8.0$, $43.39\%$ of the representation elements are selected. In addition, as shown in the averaged masks along the channel axis, the proposed method uses more representations in the high-complexity region. Over the whole Kodak image set [28], the average proportions of selected elements for the target quality levels from 1.0 to 8.0 are $6.41\%$, $9.66\%$, $14.17\%$, $19.90\%$, $27.00\%$, $35.68\%$, $46.20\%$, and $55.81\%$, respectively, where they are almost linearly proportional to the average bpp values, as shown in Figure 4.

Figure 5 shows how many representations in a low target quality level are commonly used (or selected) for higher target quality levels. For example, the orange line indicates that $100\%$, $99.8\%$, $99.6\%$, $99.0\%$, $98.3\%$ and $98.2\%$ of the representation elements, which are selected for a target quality level $q = 2.0$, are also reused for higher target quality levels from $q = 3.0$ to $8.0$, respectively. It should be noted in Figure 5 that the case with $q = 8.0$ tends to highly reuse $97.6\%$ of the selected representation elements for the case with $q = 1.0$. This implies that the proposed SCR method does not select the representation elements separately for different target quality levels, but actively takes a large portion of representation elements as common components for various target quality levels.

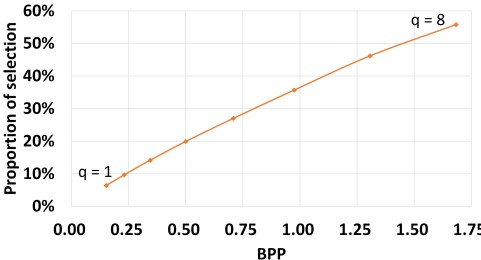

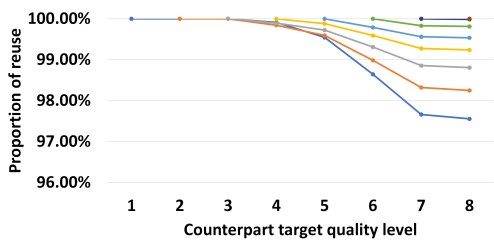

Figure 4: Average proportions of selected representation elements versus average BPP. (test set: the Kodak image set [28], base model: Hyperprior [6])

Figure 5: Average proportions of reused representation elements from low to high quality levels. (test set: the Kodak image set [28], base model: Hyperprior [6])

### 3.3 Training

We train the proposed SCR model in an end-to-end manner, where the base compression model, 3D importance map generation module, $\boldsymbol{\gamma}_q$ vectors, $\boldsymbol{QV}_q$ vectors, and $\boldsymbol{IQV}_q$ vectors are optimized all together, using the total loss formulated as follows:

$$\mathcal{L} = \sum_q R_q + \lambda_q * D_q, \ \ \text{with } R_q = H_q(\tilde{\boldsymbol{y}}_q^s \mid \tilde{\boldsymbol{z}}) + H(\tilde{\boldsymbol{z}}), \tag{4}$$

where $R_q$ and $D_q$ represent rate and distortion terms, respectively, for the target quality level $q$, and $\lambda_q = 0.2 \cdot 2^{q-8}$ is a parameter for adjusting the balance between the rate and distortion. $D_q$ can be either the mean squared error (MSE) or MS-SSIM between the input image $\boldsymbol{x}$ and the reconstructed image $\boldsymbol{x}_q'$. For the MS-SSIM-based optimization, we use the distortion term $D_q = 3000(1 - \textit{MS-SSIM}(\boldsymbol{x}, \boldsymbol{x}_q'))$. $H(\cdot)$ is a calculated cross-entropy for the quantized representations of $\boldsymbol{y}$ and $\boldsymbol{z}$. In the case of $\boldsymbol{y}$, because the quantization and mask generation processes are different for each target quality level $q$, we use a cross entropy $H_q(\cdot)$ for a target quality level $q$ as follows:

$$H_q(\tilde{\boldsymbol{y}}_q^s \mid \tilde{\boldsymbol{z}}) = \frac{1}{N_{\boldsymbol{x}}} \sum_{i=1}^{N_q^s} -\log_2 P_q(\tilde{y}_{q,i}^s | \hat{\boldsymbol{z}}), \tag{5}$$

$$\text{with } \tilde{\boldsymbol{y}}_q^s = M(\boldsymbol{y}/\boldsymbol{QV}_q + U(-0.5, 0.5), \tilde{m}(\hat{\boldsymbol{z}}, q)),$$

where $N_{\boldsymbol{x}}$ is the number of pixels in a input image $\boldsymbol{x}$, $N_q^s$ is a total number of the selected elements $\{\tilde{y}_{q,i}^s\}_{i=1}^{N_q^s}$ of $\tilde{\boldsymbol{y}}_q^s$. The cross entropy, $H_q(\tilde{\boldsymbol{y}}_q^s \mid \tilde{\boldsymbol{z}})$, of the selected representation elements is calculated based on an approximate probability mass function (PMF) $P_q(\cdot)$ to deal with the distributions of $\tilde{\boldsymbol{y}}_q^s$ that vary for different target quality levels. Specifically, we determine the estimated distribution parameters $\boldsymbol{\mu}_q$ and $\boldsymbol{\sigma}_q$ of $P_q(\cdot)$ as $M(\boldsymbol{\mu}/\boldsymbol{QV}_q, m(\hat{\boldsymbol{z}}, q))$ and $M(\boldsymbol{\sigma}/\boldsymbol{QV}_q, m(\hat{\boldsymbol{z}}, q))$, respectively. Here, the $\boldsymbol{\mu}$ and $\boldsymbol{\sigma}$ values are obtained from the base compression models. For the Context [7] base model, the position-wise parameters $\boldsymbol{\mu}_q^{(k,l)}$ and $\boldsymbol{\sigma}_q^{(k,l)}$ are obtained for each spatial coordinate $(k, l)$ through $M(\boldsymbol{\mu}^{(k,l)}/\boldsymbol{QV}_q, m(\hat{\boldsymbol{z}}, q)^{(k,l)})$ and $M(\boldsymbol{\sigma}^{(k,l)}/\boldsymbol{QV}_q, m(\hat{\boldsymbol{z}}, q)^{(k,l)})$, respectively. Note that we disregard $\boldsymbol{\mu}_q$ when a zero-mean Gaussian-based model is used for $P_q(\cdot)$. As in the previous entropy minimization-based compression models [6, 7], we adopt a Gaussian distribution model convolved with a uniform distribution as an approximate PMF $P_q(\cdot)$, and use the representation with additive uniform noise $U(-0.5, 0.5)$, denoted as $\tilde{\boldsymbol{y}}_q^s$, for training, rather than the rounded representation $\hat{\boldsymbol{y}}_q^s$ for inference. To handle the instability in the training phase due to learning the binary representations of the mask, we use a stochastically generated mask $\tilde{m}(\cdot)$ rather than $m(\cdot)$ used in the test phase. While the adjusted 3D importance map is simply rounded-off for $m(\cdot)$, $\tilde{m}(\cdot)$ is constructed with randomly sampled binary representations, similarly to Raiko *et al.* [29]'s approach, by regarding each element value of the adjusted 3D importance map $im(\hat{\boldsymbol{z}})^{\gamma_q}$ as the probability that the corresponding component of the output mask is "1", which is given as follows:

$$\tilde{m}(\hat{\boldsymbol{z}}, q) = B(im(\hat{\boldsymbol{z}})^{\gamma_q} + U(-0.5, 0.5)) \tag{6}$$

Note that discontinuity caused by the rounding-off operator $B(\cdot)$ is handled by bypassing the gradients backward. In the actual implementation, the training can be performed without using $M(\cdot)$ and $Re(\cdot)$, because we can exclude the unselected representations using Eq. 7 to calculate $R_q$, and can obtain $\breve{\boldsymbol{y}}_q$ via $AdaQ_q(\boldsymbol{y}) \cdot \tilde{m}(\hat{\boldsymbol{z}}, q)$ to compute $D_q$. Other training details are described in Appendix B.

$$H_q(\tilde{\boldsymbol{y}}_q^s \mid \tilde{\boldsymbol{z}}) = \frac{1}{N_{\boldsymbol{x}}} \sum_i -\log_2 P_q(\tilde{y}_{q,i}) \cdot \tilde{m}(\hat{\boldsymbol{z}}, q)_i, \ \ \text{with } \tilde{\boldsymbol{y}}_q = \boldsymbol{y}/\boldsymbol{QV}_q + U(-0.5, 0.5) \tag{7}$$

## 3.4 Continuously variable-rate compression

To support the continuously variable-rate compression during the test, we determine $\boldsymbol{\gamma}_q$ by interpolation when $q$ is a value between two discrete target quality levels as follows:

$$\boldsymbol{\gamma}_q = \begin{cases} \boldsymbol{\gamma}_q, & \text{if } q \in \{1, 2, ..., N_Q\} \\ \boldsymbol{\gamma}_{\lfloor q \rfloor}^{1-(q-\lfloor q \rfloor)} \cdot \boldsymbol{\gamma}_{\lceil q \rceil}^{q-\lfloor q \rfloor}, & \text{otherwise} \end{cases} \tag{8}$$

For example, when $q$ is 3.8, $\boldsymbol{\gamma}_{3.8}$ is determined by the element-wise multiplication of $\boldsymbol{\gamma}_{3.0}^{0.2}$ and $\boldsymbol{\gamma}_{4.0}^{0.8}$. Note that the $\boldsymbol{QV}_q$ and $\boldsymbol{IQV}_q$ vectors are also interpolated in the same manner as above, and this non-linear interpolation is inspired by Cui *et al.* [19]. The experimental results of the continuously variable-rate are provided in Sec. 4.1.

# 4 Experiments

## 4.1 Coding efficiency measurement for various target quality levels

**Experiments for discrete target quality levels**. To show the effectiveness of our SCR method with selective compression and adaptive quantization, we integrate it into the following three reference compression models: Hyperprior [6], Mean-scale [7] and Context [7] which are denoted as $SCR_{Hyp}$, $SCR_{MS}$ and $SCR_{Cxt}$, respectively. These three extended models with our SCR method are compared with their original models in terms of PSNR (MS-SSIM) BD-rate. The BD-rate values are measured with the target quality levels of q = 1.0, 4.0, 6.0 and 8.0 of our SCR models and the corresponding four compression points of the original models. The experimental results of ablation study are also provided to show the effectiveness of our selective compression where the $SCR_{Hyp}$, $SCR_{MS}$ and $SCR_{Cxt}$ are compared with and without the selective compression modules. Here, the SCR variant without the selective compression is equivalent to the adaptive quantization-based variable-rate compression method of Cui *et al.* [19], where Gain and Inverse-gain vectors are used, corresponding to the $QV_q$ and $IQV_q$ vectors in our work, respectively. Note that we additionally apply the adaptive $\boldsymbol{\mu}$ and $\boldsymbol{\sigma}$ adjustment process described in Section 3.3 for each target quality level. For comparisons, we train each of the $SCR_{Hyp}$, $SCR_{MS}$ and $SCR_{Cxt}$ as two different versions, an MSE(PSNR)-optimized model and an MS-SSIM-optimized model which are denoted as $SCR_*^{psnr}$ and $SCR_*^{ssim}$, respectively, where $*$ represents the used reference model. We measure average bpp-PSNR and bpp-MS-SSIM values over the Kodak image set [28].

As shown in Table 1, our SCR models achieve comparable results to all the reference models separately trained for various target quality levels. In addition, compared with $SCR_{Hyp}^{psnr}$, $SCR_{MS}^{psnr}$, and $SCR_{Cxt}^{psnr}$ without the selective compression, our SCR (full) models obtained coding efficiency gains of $-4.30\%$, $-5.03\%$, and $-1.18\%$ in BD-rate, respectively. This clearly demonstrates that the selective compression in the SCR method is effective in terms of the coding efficiency. For the MS-SSIM-optimized models, we obtained similar results as those of MSE-optimized models. The rate-distortion curve for $SCR_{Cxt}^{psnr}$ and the MS-SSIM-optimized models are provided in Appendix D. As shown in Figure 6, the coding efficiency gains of our SCR models are more noticeable for the low bit rate, compared with the SCR variants without selective compression. This might be because the SCR (w/o selective compression) models have to encode all the representation elements regardless of target quality levels, although some representation elements may need to be optimized only for a certain bit rate. Whereas, the SCR (full) model can effectively exclude those unnecessary representation elements, especially at low-bpp range, thus leading to more efficient compression.

For the Context [7] reference model, the $SCR_{Cxt}^{psnr}$ and $SCR_{Cxt}^{ssim}$ models obtained the relatively smaller coding efficiency gains over their variants without the selective compression. This may come from two reasons: (i) In the autoregressive Context [7] model, the reconstructed representation elements are utilized to predict the distribution parameters of the adjacent elements. Thus, in $SCR_{Cxt}$ models, the unselected representations filled with 0 values may deteriorate the prediction accuracy; (ii) Unlike for the other two reference compression models, Hyperprior [6] and Mean-scale [7], the $SCR_{Cxt}$ variants without the selective compression already achieved a fairly satisfactory coding efficiency with respect to the original Context [7] models, so there seems to be not much room for performance improvement. Although there exist somewhat deviations in coding efficiency gain for several reference models, it is worthwhile to note that our SCR models achieve comparable results to all the reference models over the wide range of bit rates, as shown in Figure 6 and Appendix D.

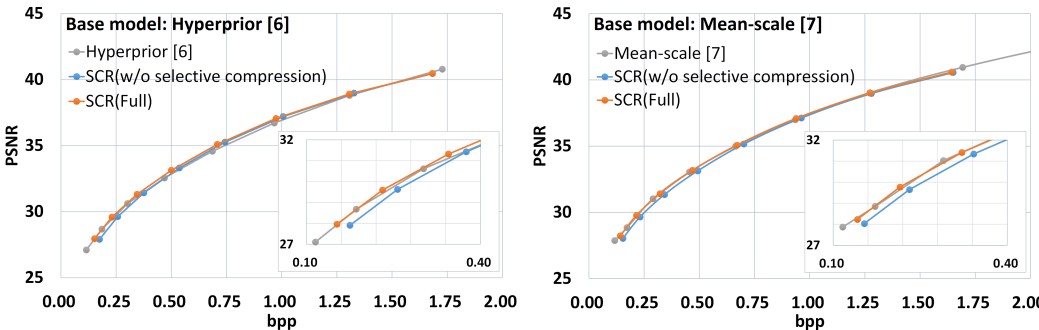

Figure 6: Rate-distortion curves in discrete target quality levels for the reference compression model, the SCR variants without the selective compression, and SCR full model. Here, MSE-optimized models are used and the results for the MS-SSIM-optimized models are provided in Appendix D.

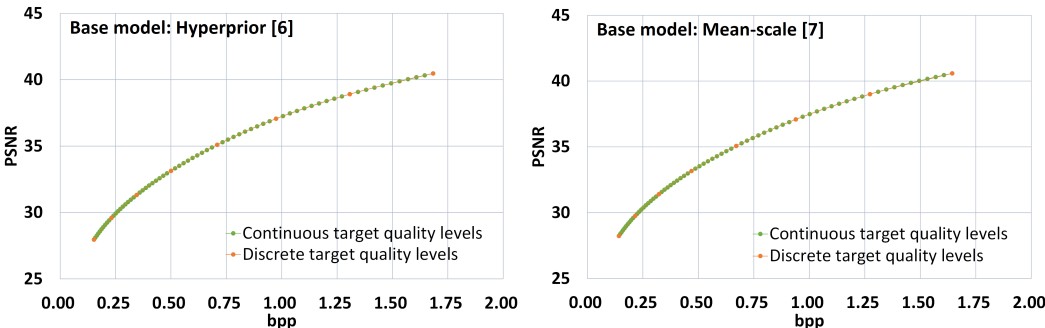

Figure 7: Rate-distortion curves of our SCR model in continuous target quality levels. The orange dots indicate the results for discrete target quality levels while the green dots represent the results using $\gamma_q$ vector interpolation.

In addition, we reproduce some prior variable-rate compression methods [19, 20, 23, 24, 25] on the Hyperprior [6] architecture and compare them with our SCR model in terms of coding efficiency and decoding time. As described in Appendix F, our SCR model shows superior results compared with those models. The experimental results and further discussions are provided in Appendix F.

**Experiments for fine-grain target quality levels**. To verify the continuously variable-rate compression of the proposed SCR method, we measure the rate-distortion performance by changing $q$ from 1.0 to 8.0 with an increment of 0.1. As shown in Figure 7, the proposed SCR method stably supports the continuously variable-rate compression without degrading coding efficiency. Figure 8 shows the reconstructed samples of the proposed SCR method where the quality gradually improves as $q$ increases. More reconstruction samples are provided in Appendix G.

Table 1: BD-rate results of our SCR models compared with the corresponding reference compression models (top three rows) and the SCR variants without selective compression (bottom three rows) denoted as "w/o SC". A negative BD-rate value means a coding efficiency gain.

| MSE-optimized | | | MS-SSIM-optimized | | |
|---|---|---|---|---|---|
| Ref. | Tested | BD-rate | Ref. | Tested | BD-rate |
| Hyperprior [6] | $SCR_{Hyp}^{psnr}$ | **-3.21%** | Hyperprior [6] | $SCR_{Hyp}^{ssim}$ | **-0.91%** |
| Mean-scale [7] | $SCR_{MS}^{psnr}$ | 1.30% | Mean-scale [7] | $SCR_{MS}^{ssim}$ | 0.96% |
| Context [7] | $SCR_{Cxt}^{psnr}$ | 1.23% | Context [7] | $SCR_{Cxt}^{ssim}$ | **-2.00%** |
| $SCR_{Hyp}^{psnr}$ (w/o SC) | $SCR_{Hyp}^{psnr}$ | **-4.30%** | $SCR_{Hyp}^{ssim}$ (w/o SC) | $SCR_{Hyp}^{ssim}$ | **-4.03%** |
| $SCR_{MS}^{psnr}$ (w/o SC) | $SCR_{MS}^{psnr}$ | **-5.03%** | $SCR_{MS}^{ssim}$ (w/o SC) | $SCR_{MS}^{ssim}$ | **-3.00%** |
| $SCR_{Cxt}^{psnr}$ (w/o SC) | $SCR_{Cxt}^{psnr}$ | **-1.18%** | $SCR_{Cxt}^{ssim}$ (w/o SC) | $SCR_{Cxt}^{ssim}$ | **-1.22%** |

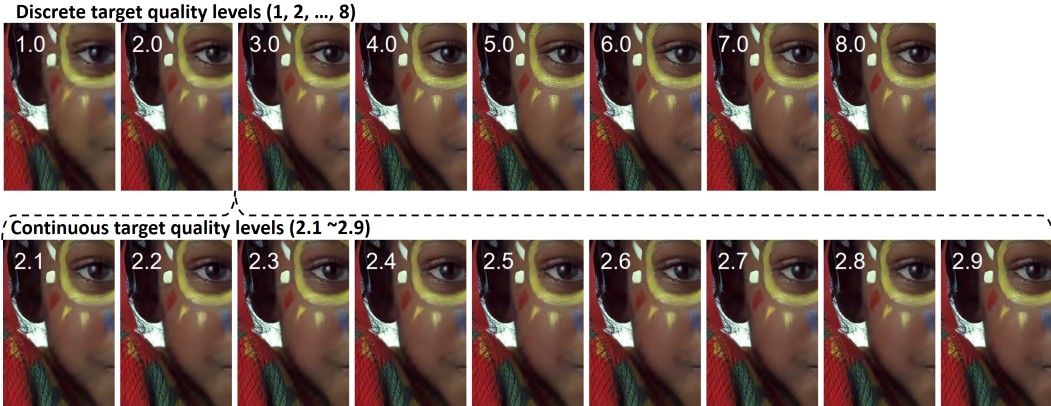

Figure 8: Qualitative results of the proposed SCR method (best viewed in digital format). For best viewing, we cropped the reconstruction results. The number embedded in each sample represents the target quality level $q$. The top row represents the reconstruction results of the discrete quality levels. The bottom row shows the reconstruction results between $q = 2.0$ and $q = 3.0$ with the interpolation method in Sec. 3.4.

## 4.2 Decoding time

We compare the decoding times for the original reference compression models (Hyperprior [6] and Mean-scale [7]) and their SCR extensions ($SCR_{Hyp}^{psnr}$, $SCR_{MS}^{psnr}$) for variable bit rate image compression. To show the effect of the selective compression in our SCR method on the decoding time, the decoding times of the above models, $SCR_{Hyp}^{psnr}$ and $SCR_{MS}^{psnr}$, are compared for the cases with and without the selective compression in the same target quality levels. For the comparison of the two reference compression models, Hyper [6] and Mean-scale [7], their decoding times are measured for a high quality compression model that corresponds to our variable-rate configuration with $q = 8.0$. Note that, since each of the two reference compression models has different architectures for different target quality levels, the models corresponding to the target quality level of $q = 8.0$ are selected for comparison, which have the same en/decoder network architectures as those of $SCR_{Hyp}^{psnr}$ and $SCR_{MS}^{psnr}$. We perform decoding 100 times for each image in the Kodak image set [28] and measure the average decoding time of all the images. It should be noted that a GPU is used for the decoder network processing, whereas we use a CPU for hyper-decoder processing to prevent the encoder and decoder discrepancy owing to the GPU characteristic that incurs tiny errors for the same inputs even on the same machine, because of its parallel algorithm that yields different numeric results [30].

As shown in Figure 9, $SCR_{Hyp}^{psnr}$ and $SCR_{MS}^{psnr}$ reduce the decoding time in the entire bit rate range compared with their variants without the selective compression. The decoding time savings are more noticeable for the low-quality compression owing to the reduction in the number of selected representation elements, and this is clearly seen when comparing the entropy decoding times (denoted in light blue color). In spite of the overheads from the reshaping (in dark blue) and 3D binary mask generation, the time saved in the entropy decoding process is definitely larger than the overhead time. It should be noted that the 3D binary mask generation time is included in the hyper decoder network time (gray) because the 3D binary mask generation shares most parts of the hyper decoder. $SCR_{Hyp}^{psnr}$ and $SCR_{MS}^{psnr}$ reduce the average decoding time by $11.28\%$ and $8.32\%$, respectively, compared with their variants without the selective compression. Furthermore, $SCR_{Hyp}^{psnr}$ and $SCR_{MS}^{psnr}$ show lower decoding times, compared with their corresponding reference compression models with $q = 8.0$, over all target quality levels except for $q = 8.0$ of $SCR_{MS}^{psnr}$. Instead, $SCR_{MS}^{psnr}$ with $q = 8.0$ yielded a slightly higher decoding time compared with the original Mean-scale [7] model ($q = 8.0$).

For Context [7]-based models, the $SCR_{Cxt}^{psnr}$ reduces the average decoding time by $25.71\%$ and $21.53\%$ compared with the its variants without the selective compression and the reference Context [7] model (corresponding to q=8.0), respectively. These results clearly show the effectiveness of selective compression, but it should be noted that we use a different entropy coder for $SCR_{Cxt}^{psnr}$ from those for $SCR_{Hyp}^{psnr}$ and $SCR_{MS}^{psnr}$. The detailed information on the entropy coders is provided in Appendix C. Regarding the encoding time, it is also significantly reduced due to selective compression as done for the decoder. The encoding time results are provided in Appendix E.

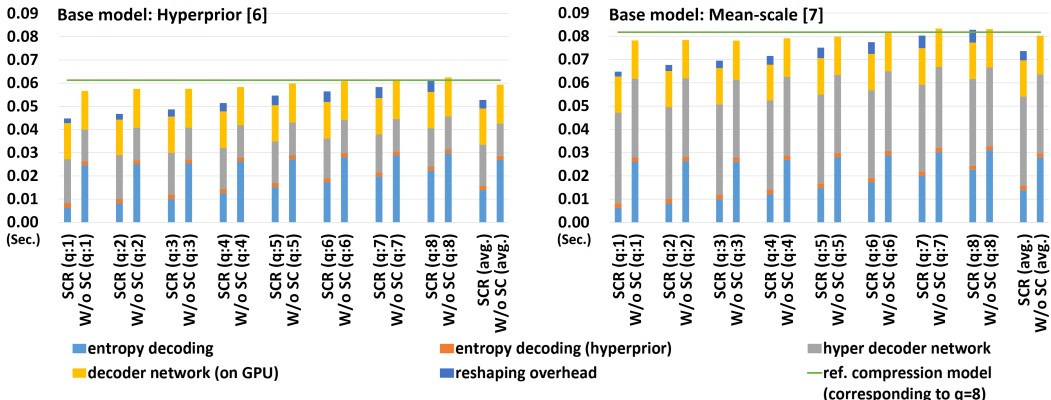

Figure 9: Decoding times of the proposed SCR models, the SCR variants without the selective compression, denoted as "W/o SC", and the reference compression models. We measured the average decoding time of each model over the Kodak image dataset [28] with 2×Intel Xeon Gold 5122 CPUs and 1×RTX Titan GPU.

Table 2: Configuration of the parameters for the $SCR_{Hyp}$ model

| Name | Description | No. of parameters |
|------|-------------|-------------------|
| Hyperprior | Original model | 11,813,443 |
| QV vectors | 8 levels * 320 ch. | 2,560 |
| IQV vectors | 8 levels * 320 ch. | 2,560 |
| gamma vectors | 8 levels * 320 ch. | 2,560 |
| Importance map generation | 192 (in) * 320 (out) + 320 (out) | 61,760 |
| Total no. of parameters | | 11,882,883 |

## 4.3 Increasing parameters due to SCR

The increase in model parameters due to the proposed SCR method is relatively very small. For example, the Hyperprior model has $11,813,443$ parameters, and the $SCR_{Hyp}$ model has $11,882,883$. That is, the number of parameters is increased by only $0.59\%$, resulting in a memory increase of $0.26$MB. Similarly, the additional memory occupancy for the $SCR_{MS}$ and $SCR_{Cxt}$ model is only $0.62$MB. The configuration of the parameters for the $SCR_{Hyp}$ model is shown in Table 2.

## 5 Conclusion

We firstly proposed a 'selective compression of representations' (SCR) method for NN-based variable-rate image compression, which performs entropy coding only for the partially selected latent representations. The proposed SCR method selects essential representation elements for compression in an adaptive manner according to a given target quality level. For this, we first generate a 3D importance map, which represents the importance of each representation element independently of target quality levels, and then adjust it in a target quality-adaptive manner using the learned importance adjustment curves to generate a 3D binary mask that represents whether or not to select each representation element for compression. We demonstrated through experiments that the proposed SCR method could provide comparable coding efficiency to those of the separately trained reference compression models. Furthermore, the proposed SCR method showed less decoding time than those under comparison for almost all the cases. We also showed that the proposed SCR method could enable continuously variable-rate image compression through simple non-linear interpolation of the importance adjustment curves between discrete target quality levels in which the selective compression was trained.

## Acknowledgments and Disclosure of Funding

This work was supported by Institute for Information & communications Technology Planning & Evaluation (IITP) grant funded by the Korea government(MSIT) (No. 2017-0-00072, Development of Audio/Video Coding and Light Field Media Fundamental Technologies for Ultra Realistic Tera-media).

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
