# OpenReview forum: "Selective compression learning of latent representations for variable-rate image compression"
_NeurIPS.cc/2022/Conference — NeurIPS 2022 Accept_

### Official Review · Reviewer_SmBv · 2022-07-10

**Rating:** 4
**Confidence:** 4
**Soundness:** 3 good
**Presentation:** 3 good
**Contribution:** 2 fair

**Summary:**

The paper proposes a variable rate image compression method using selective compression and adaptive quantization. It is shown that the RD performance is similar to that of the reference models and decoding time can be reduced.

**Questions:**

- Are QV_q and IQV_q trained together with the model?

- Is there any performance gain if the network for 3D importance map generation is more complex than a single 1x1 conv layer? Actually, [Li'18] used such a complex model.

- Just out of curiosity, what happens if q<1 or q>8?

- It is quite surprising that even if q=8, only 55.81% of the features are used. This means that about a half of the extracted features are useless anyway. Some discussion on this could be interesting. There are some studies on pruning of image compression models, which could be supported by this observation.

- In Fig. 6, the difference between SCR(Full) and SCR(w/o selective compression) becomes larger at lower bpp values. Some discussion on why this happens would be helpful.

- In Fig. 5, not all features for a value of q are used for larger values of q. This is unclear. Since the adjustment curves are monotonic, it has to be that increasing q only adds additional feature elements for encoding so the reuse rate is always 100%.

- (minor) Ripple [20] -> Rippel [20]

- (minor) Does "BD-rate gain of -3.21%" mean a gain by 3.21%? or a negative gain (i.e., loss) by 3.21%? I assume the former, but it's confusing.

**Limitations:**

Limitations are discussed in the paper.

**Strengths And Weaknesses:**

*** Strength

+ An important task, variable-rate image compression, is addressed.

+ The paper is written relatively clearly.

+ From the experiments, the method seems to work well.

*** Weakness

- The proposed method is composed of adaptive quantization and importance map-based selective compression.   First, it is said that the adaptive quantization is similar to [19,23]. It would be better to explain more concretely how similar/different the adaptive quantization in this paper is from those in [19,23]. Second, the idea of 3D importance map already exists [Li'18], although it was not used for variable rate compression. Furthermore, the added value of the selective compression seems to be limited, especially for advanced base models like the context model [7]. In fact, it will be necessary to show the effectiveness of the proposed method on more recent models like [11, 13, He'21, Gao'21, Qian'21].


[Li'18] Li et al., "Learning convolutional networks for content-weighted image compression," CVPR 2018

[He'21] He et al., "Checkerboard context model for efficient learned image compression," CVPR 2021

[Gao'21] Gao et al., "Neural image compression via attentional multi-scale back projection and frequency decomposition," ICCV 2021

[Qian'21] Qian et al., "Learning accurate entropy model with global reference for image compression," ICLR 2021

- Performance comparison with existing variable rate compression methods is missing. The proposed method needs to be compared against different ways to accomplish variable rate encoding in the previous studies.

- It is quite difficult to follow the quantitative BD-rate gain/loss results given in the text in Section 4.1. It is suggested to summarize them as a table.

---

> ### Author Response · Authors · 2022-08-02
> **Response to Reviewer 4 (SmBv)'s comments  (3/3)**
>
> • In Fig. 6, the difference between SCR(Full) and SCR(w/o selective compression) becomes larger at lower bpp values. Some discussion on why this happens would be helpful.
>
> **[Authors Reply 4-9]** Thanks for the comment. The SCR (w/o selective compression) model has to compress and reconstruct all representations, but some representations (features) for the variable-rate compression models are optimized for a certain bit rate. Particularly, some representations may be optimized for high-quality compression, while some other representations for low-quality compression. The SCR(w/o selective compression) may increases the quantization step to reduce bit-rate allocation for unnecessary representations, but those unnecessary representations eventually occupy some bit-rate more or less. On the other hand, the SCR(full) model can effectively exclude those unnecessary representations, especially at low-bpp range, thus leading to more effective pruning of the unnecessary representation components.
>
> ---
> • In Fig. 5, not all features for a value of q are used for larger values of q. This is unclear. Since the adjustment curves are monotonic, it has to be that increasing q only adds additional feature elements for encoding so the reuse rate is always 100%.
>
> **[Authors Reply 4-10]** Thanks for the comment. The adjustment curves in Fig. 2 are illustrated using the mean values of gamma_q vectors, to show an overall trend in an average sense. However, each gamma_q vector consists of C_y(the number of channels in y) components, and some of those C_y curves may not be monotonic. This is because, as replied in [Authors Reply 4-9], some representations are optimized in use only for a lower bit rate and can be ignored or deemphasized in a higher bpp range. In this case, the importance values of these representations are attenuated for the higher target quality levels.
>
> ---
> • (minor) Ripple [20] -> Rippel [20]
>
> **[Authors Reply 4-11]** Thanks for the comments. We’ll correct it in the revised manuscript.
>
> ---
> • (minor) Does "BD-rate gain of -3.21%" mean a gain by 3.21%? or a negative gain (i.e., loss) by 3.21%? I assume the former, but it's confusing.
>
> **[Authors Reply 4-12]** Thanks for the comments. Both negative and positive expressions tend to be interchangeably used for gain in the literature. The authors feel that the negative (positive) expression for gain (loss) seems to be more frequently used. We'll clarify this in the revised manuscript.
>
> ---
> [Li'18] Li et al., "Learning convolutional networks for content-weighted image compression," CVPR 2018
>
> [He'21] He et al., "Checkerboard context model for efficient learned image compression," CVPR 2021
>
> [Gao'21] Gao et al., "Neural image compression via attentional multi-scale back projection and frequency decomposition," ICCV 2021
>
> [Qian'21] Qian et al., "Learning accurate entropy model with global reference for image compression," ICLR 2021
>
> [Zhou'20] Zhou et al. "Variable Rate Image Compression Method with Dead-zone Quantizer", CVPR 2020 Workshop

---

> ### Author Response · Authors · 2022-08-02
> **Response to Reviewer 4 (SmBv)'s comments (2/3)**
>
> • Performance comparison with existing variable rate compression methods is missing. The proposed method needs to be compared against different ways to accomplish variable rate encoding in the previous studies.
>
> **[Authors Reply 4-3]** Thanks for the comment. We actually compared the performance of our SCR method with two variable rate compression methods of Cui’ [19] and Chen’ [23], as mentioned in [Authors Reply 3-1]. We’ll include more comparison results in the revised manuscript if possible. For the others [1, 17, 18, 20, 21, 22], as we mentioned in lines 235-238, they use their own base architectures or dedicated entropy model schemes, which are not a widely used backbone architecture that our SCR method is based on,  so it was not able to make direct comparisons.
>
> ---
> • It is quite difficult to follow the quantitative BD-rate gain/loss results given in the text in Section 4.1. It is suggested to summarize them as a table.
>
> **[Authors Reply 4-4]** Thanks for the comment. To reflect the reviewer’s comment, we’ll summarize the results in a new table as below:
>
> ---
> **<MSE-optimized models>**
> |Ref. model|Tested model|BD-rate (the smaller the better)|
> |---|---|---|
> |Hyper|$SCR_{Hyp}^{psnr}$|-3.21%|
> |$SCR_{Hyp}^{psnr}$ (w/o SC)|$SCR_{Hyp}^{psnr}$|-4.30%|
> |Mean-scale|$SCR_{MS}^{psnr}$|1.30%|
> |$SCR_{MS}^{psnr}$ (w/o SC)|$SCR_{MS}^{psnr}$|-5.03%|
> |Context|$SCR_{Cxt}^{psnr}$|1.23%|
> |$SCR_{Cxt}^{psnr}$ (w/o SC)|$SCR_{Cxt}^{psnr}$|-1.18%|
> ---
> **<MS-SSIM-optimized models>**
> |Ref. model|Tested model|BD-rate (the smaller the better)|
> |---|---|---|
> |Hyper|$SCR_{Hyp}^{ssim}$|-0.91%|
> |$SCR_{Hyp}^{ssim}$ (w/o SC)|$SCR_{Hyp}^{ssim}$|-4.03%|
> |Mean-scale|$SCR_{MS}^{ssim}$|0.96%|
> |$SCR_{MS}^{ssim}$ (w/o SC)|$SCR_{MS}^{ssim}$|-3.00%|
> |Context|$SCR_{Cxt}^{ssim}$|-2.00%|
> |$SCR_{Cxt}^{ssim}$ (w/o SC)|$SCR_{Cxt}^{ssim}$|-1.22%|
>
>
> ## Questions:
> ---
> • Are QV_q and IQV_q trained together with the model?
>
> **[Authors Reply 4-5]** Thanks for the comment. Yes. QV_q and IQV_q vectors are optimized together with other parameters. We’ll more clearly specify this in Section 3.3.
>
> ---
> • Is there any performance gain if the network for 3D importance map generation is more complex than a single 1x1 conv layer? Actually, [Li'18] used such a complex model.
>
> **[Authors Reply 4-6]** Thanks for the comment. We tried with a bit deeper network module than one single 1x1 conv. layer, but obtained almost the same coding efficiency. This is because not only the one single 1x1 conv. layer is involved but also the hyper encoder/decoder heavily get involved the 3D importance map compression/reconstruction (generation) functions together with their original functions.
> Regarding the work [Li’18], they used a 2D importance map, not a 3D importance map, that represents only the spatial importance. Thus, we cannot use their method to represent the component-wise importance values.
>
> ---
> • Just out of curiosity, what happens if q<1 or q>8?
>
> **[Authors Reply 4-7]** Thanks for the comment. Our SCR method can generically cover a wide range of bit rates. For q>8,  the number of hyperparameters (the sizes of en/decoder) may need to be increased to represent finer details of the input images. For q<1, we expect that the current SCR models can be extended to the low bit-rate range.
>
> ---
> • It is quite surprising that even if q=8, only 55.81% of the features are used. This means that about a half of the extracted features are useless anyway. Some discussion on this could be interesting. There are some studies on pruning of image compression models, which could be supported by this observation.
>
> **[Authors Reply 4-8]** Thanks for the comment. The number of feature channels used in the base model is 320 where an average of 55.81% of feature elements of the channels are regarded as essential representations for q = 8.0. This is exactly the advantage of our selective compression method as described in Section 3.1 of the manuscript. It should be noted that our selective compression method can be regarded as component-wise pruning which should be distinguished from the traditional channel-wise pruning of CNN. Also, our selective compression method performs component-wise pruning in the RD sense while the traditional channel-wise pruning of CNN, for example, may be done in the perspective of L1 or L2 loss optimization.

---

> ### Author Response · Authors · 2022-08-02
> **Response to Reviewer 4 (SmBv)'s comments (1/3)**
>
> **[Authors]** First of all, we really appreciate your careful comments. We address comments as below:
>
> ## Weakness
> ---
> • The proposed method is composed of adaptive quantization and importance map-based selective compression. First, it is said that the adaptive quantization is similar to [19,23]. It would be better to explain more concretely how similar/different the adaptive quantization in this paper is from those in [19,23].
>
> **[Authors Reply 4-1]** Thanks for the comment. As described in line 234, “the SCR variant without the selective compression” is the variable-rate compression model only using the adaptive quantization based on QV_q and IQV_q vectors. We applied the mu and sigma adjustment process for each target quality level, which is not described in [19]. In [23], they also use the scaling and inverse scaling factors for representations, and they additionally utilize the shifting factors. To reflect the reviewer’s comment, we’ll more clearly explain this in the revised manuscript.
>
> ---
> • Second, the idea of 3D importance map already exists [Li'18], although it was not used for variable rate compression. Furthermore, the added value of the selective compression seems to be limited, especially for advanced base models like the context model [7]. In fact, it will be necessary to show the effectiveness of the proposed method on more recent models like [11, 13, He'21, Gao'21, Qian'21].
>
> **[Authors Reply 4-2]** Thanks for the comment. The mentioned 3D importance map for [Li'18] is not 3D but a 2D importance map which is used for a spatially different allocation of bits. It is important to note that the mentioned prior work [Li'18] with 2D importance maps is completely different from our 3D importance maps in perspectives of usage purpose and operations, which is not just a matter of 2D versus 3D importance maps. Please refer to [Authors Reply 2-2].
>
> Regarding the comment with a limited effect of the selective compression method on the context model, it is not really the point of the effectiveness for the selective compression method. As mentioned in [Authors Reply 3-2 and 3-4], the coding efficiencies of single fixed-rate models trained on their specific target bit rates are the upper performance bound for variable-rate compression models. The main goal for the variable-rate compression models is to achieve comparable coding efficiency as the single fixed-rate models over a wide range of bit rates. Actually, our SCR method shows comparable coding efficiency performance with Context model [R3.2]. As shown in Fig. 11 of the Supplemental, our SCR method is comparable with those of the single models with Context models [R3.2] even over a wide range of bit rates, from 0.1 to 1.75bpp (PSNR 28dB ~ over 40dB), which is much wider(x2) range compared to bit rate range supported by the previous variable-rate compression models [19, 23].
>
> Regarding the suggested works with more recent models like [11, 13, He'21, Gao'21, Qian'21], our experiments were carried out with three popular base compression architecture (Hyperprior, Mean-scale, Context). As the reviewer might know, the source codes are not usually available in the learned compression areas, not like in computer vision deep learning areas. To make it worse, in the learned image compression domain, it is really difficult to reproduce the same results as the methods described in the original papers because there exist many implementation-specific details such as hyper parameters or tune values that are not described in the papers, which may result in very different R-D performances compared to those reported in the papers. We tested our SCR method with three most popular base compression architectures (Hyperprior, Mean-scale, Context) from which our SCR method has shown comparable coding efficiency to the upper performance bounds with their respective single-rate models. Please note that very recent works such as Cui [19] and Chen [23] used three baseline models. Another variable-rate compression model of Zhou [Zhou'20] used one baseline model.
>
> As another big advantage, as replied in [Authors Reply 3-2], our SCR method also significantly reduces the en/decoding time of variable rate compression on the Context models, while providing comparable coding efficiency to the separately trained Context models. Specifically, the $SCR_{Cxt}^{psnr}$ (full) model reduces the decoding time by 25.71% compared with the $SCR_{Cxt}^{psnr}$ (w/o SC) model, and by 21.53% compared with the ref. Context model (corresponding to q=8.0). In the case of encoding, the $SCR_{Cxt}^{psnr}$ (full) model reduces the decoding time by 20.25% compared with the $SCR_{Cxt}^{psnr}$ (w/o SC) model, and by 13.90% compared with the Context model (corresponding to q=8.0).

---

> ### Author Response · Authors · 2022-08-09
> **Additional response to Reviewer 4 (SmBv)'s comments**
>
> We received the experimental results of Chen [23] from the authors. On the Hyperprior [6] architecture, Our SCR method outperforms Chen [23] by 13.42% (1.64%) for the MS-SSIM (MSE)-optimized model in terms of MS-SSIM (PSNR) BD-rate. It should be noted that our SCR method not only outperforms Chen [23] in coding efficiency but also supports more than twice the bit-rate ranges compared to those in Chen [23], as below:
>
> <$SCR_{Hyp}$>
> - 0.209 to 1.819 bpp for the MS-SSIM-optimized model
> - 0.152 to 1.683 bpp for the MSE-optimized model
>
> <"SF in Balle (2018)" in Fig. 6 of Chen [23]>
> - 0.250 to 0.927 bpp for the MS-SSIM-optimized model
> - 0.187 to 0.837 bpp for the MSE-optimized model.
>
> For your information, in the comparison, we used the four target quality levels (q = 1.0, 3.0, 5.0, and 6.0) of our SCR models to match the bit-rate ranges as much as possible. When we used the whole range of our SCR models with q = 1.0, 4.0, 6.0, and 8.0, we obtained slightly greater BD-rate gains.
>
> We'll add these comparison results in the revised manuscript, and we would appreciate it if you could consider these comparison results as well.

---

### Official Review · Reviewer_mgum · 2022-07-11

**Rating:** 5
**Confidence:** 5
**Soundness:** 2 fair
**Presentation:** 2 fair
**Contribution:** 1 poor

**Summary:**

Currently used variable-rate image compression model usually adds additional modules to the base model which increases the complexity while those using adaptive quantization to achieve variable-rate compression face degradation in coding efficiency. To address this problem, this paper introduces selective compression of representations (SCR), which embeds selective compression into variable-rate models based on adaptive compression. The selective compression mechanism first generates a 3D importance map according to the output of the hyper decoder and then adjusts it using the channel-wise parameter , and finally, it is binarized to indicate whether the element is selected for compression. Also, this paper introduces a interpolation-based mothed to support continously variable-rate compression when the quality level q is a value between two discrete target quality levels.


**Questions:**

The two base models shown in the experiment share similar encoder/decoder structure and both use hyperpiror as the entropy model, the author should include more method with a greater diversity of structures in their experiments.

**Ethics Review Area:**

["I don’t know"]

**Limitations:**

The proposed method may not be generalized to learned image compression with context adaptive entropy engine.

**Strengths And Weaknesses:**

Strengths:
1. According to the experiment, the proposed selective compression mechanism can distinctly increase the compression efficiency when using "Hyperpiror [1]" and "Mean-scale [2]" as the base model. Also, because the proposed method only performs entropy coding for some of the elements, the decoding time decreases, especially at low bitrate. The proposed method only uses a single 1*1 convolutional layer to generate the 3D importance map and the parameter has a limited number of parameters, so it adds few parameters to the base model.
Weaknesses:
1. This article argues that variable-rate compression methods like [3] may increase the compression complexity by adding additional modules to the base model, so the author chooses to implement selective compression on the adaptive-quantization-based method. Also, the author chooses the model mentioned in [1] and [2] as the base model. However, the overhead complexity of the added module may not be that high compared to the base model itself. Also, it lacks a comparison between the proposed method and other variable-rate models which add modules to the base model in compress efficiency and complexity.
2. According to Appendix A.3, the selective compression method has little impact when implemented on "Context [2]". While the state-of-the-art learned image codecs usually adopt context model in their entropy engine, the proposed method may be unable to increase the compression efficiency when using the state-of-the-art codec like [4] as the base model.



[1] J. Ballé, D. Minnen, S. Singh, S. J. Hwang, and N. Johnston, “Variational image compression with a scale hyperprior,” in the 6th Int. Conf. on Learning Representations, 2018.
[2] D. Minnen, J. Ballé, and G. Toderici, “Joint autoregressive and hierarchical priors for learned image compression,” in Advances in Neural Information Processing Systems, May 2018.
[3] Y. Choi, M. El-Khamy, and J. Lee, “Variable rate deep image compression with a conditional autoencoder,”  in International Conference on Computer Vision, 2019.
[4] Cheng, Zhengxue, et al. "Learned image compression with discretized gaussian mixture likelihoods and attention modules." Proceedings of the IEEE/CVF Conference on Computer Vision and Pattern Recognition. 2020.

---

> ### Author Response · Authors · 2022-08-02
> **Response to Reviewer 3 (mgum)'s comments  (2/2)**
>
> • According to Appendix A.3, the selective compression method has little impact when implemented on Context [R3.2]". While the state-of-the-art learned image codecs usually adopt context model in their entropy engine, the proposed method may be unable to increase the compression efficiency when using the state-of-the-art codec like [R3.4] as the base model.
>
> **[Authors Reply 3-2]** Thanks for the comment. In Appendix A.3, it is noted that our SCR method has a relatively reduced impact on coding efficiency with Context [R3.2], not indicating “little impact” on it. Please note that the coding efficiencies of the single models trained on their specific target bit rates are the upper performance bound. In comparing the performance of our SCR  method with such upper performance bounds, it can be clearly noted in Fig. 11 of Appendix A.3 that the performance of our SCR method is comparable with those of the single models over a wide range of bit rates. To our best knowledge, our SCR method is the first work of variable-rate compression that shows such comparable performance of coding efficiency in comparison with separately trained single-rate models of Context model [2], even over a much wider (~2X) range of bit rates.  (The variable-rate compression results of [19] on Context models may not be comparable to those reported in the original paper of Context [7]. please also refer to [Authors Reply 3-1] and [Authors Reply 1-6].)
>
> In addition, we want to emphasize that our SCR method significantly reduces the en/decoding time for Context-based models. Specifically, the $SCR_{Cxt}^{psnr}$ (full) model reduces the decoding time by 25.71% compared with the $SCR_{Cxt}^{psnr}$ (w/o SC) model, and by 21.53% compared with the ref. Context model (corresponding to q=8.0). In the case of encoding, the $SCR_{Cxt}^{psnr}$ (full) model reduces the decoding time by 20.25% compared with the $SCR_{Cxt}^{psnr}$ (w/o SC) model, and by 13.90% compared with the ref. Context model (corresponding to q=8.0). Please refer to [Authors Reply 1-10].
>
> [R3.1] J. Ballé, D. Minnen, S. Singh, S. J. Hwang, and N. Johnston, “Variational image compression with a scale hyperprior,” in the 6th Int. Conf. on Learning Representations, 2018.
>
> [R3.2] D. Minnen, J. Ballé, and G. Toderici, “Joint autoregressive and hierarchical priors for learned image compression,” in Advances in Neural Information Processing Systems, May 2018.
>
> [R3.3] Y. Choi, M. El-Khamy, and J. Lee, “Variable rate deep image compression with a conditional autoencoder,” in International Conference on Computer Vision, 2019.
>
> [R3.4] Cheng, Zhengxue, et al. "Learned image compression with discretized gaussian mixture likelihoods and attention modules." Proceedings of the IEEE/CVF Conference on Computer Vision and Pattern Recognition. 2020.
>
> ## Questions:
> ---
> • The two base models shown in the experiment share similar encoder/decoder structure and both use hyperpiror as the entropy model, the author should include more method with a greater diversity of structures in their experiments.
>
> **[Authors Reply 3-3]** Thanks for the comment. Actually the three most popular base compression architectures (Hyperprior, Mean-scale, Context) were used in our experiments with our selective compression incorporated. Please note that very recent works such as Cui [19] and Chen [23] use three baseline models. Another variable-rate compression model of Zhou [Zhou'20] used one baseline model.
>
> [Zhou'20] Zhou et al. "Variable Rate Image Compression Method with Dead-zone Quantizer", CVPR 2020 Workshop
>
> ## Limitations:
> ---
> • The proposed method may not be generalized to learned image compression with context adaptive entropy engine.
>
> **[Authors Reply 3-4]** Thanks for the comment. The commented limitation is not correct. As mentioned in [Authors Reply 3-2], the coding efficiencies of the single models trained on their specific target bit rates are the upper performance bound. Our SCR method is comparable with those of the single models in Context [R3.2] even over a much wider (~2X) range of bit rates. To our best knowledge, our SCR method is the first work of variable-rate compression that shows such comparable performance of coding efficiency in comparison with separately trained single models in Context [R3.2].
>
> Furthermore, as replied in [Authors Reply 3-2], our SCR method also significantly reduces the en/decoding time of variable rate compression on the Context ref. models, while providing comparable coding efficiency to the separately trained ref. Context models. Specifically, the $SCR_{Cxt}^{psnr}$ (full) model reduces the decoding time by 25.71% compared with the $SCR_{Cxt}^{psnr}$ (w/o SC) model, and by 21.53% compared with the ref. Context model (corresponding to q=8.0). In the case of encoding, the $SCR_{Cxt}^{psnr}$ (full) model reduces the decoding time by 20.25% compared with the $SCR_{Cxt}^{psnr}$ (w/o SC) model, and by 13.90% compared with the ref. Context model (corresponding to q=8.0).

---

> > ### Comment · Reviewer_mgum · 2022-08-10
> > **Thanks for the response.**
> >
> > I think authors have given a great response. Many concerns are resolved. I incline to recommend the acceptance of this work.

---

> > > ### Author Response · Authors · 2022-08-10
> > > **Thanks for the careful review and judgment.**
> > >
> > > Thanks for your specific and comprehensive understanding of our rebuttals. We authors also appreciate your final judgment on our work.

---

> ### Author Response · Authors · 2022-08-02
> **Response to Reviewer 3 (mgum)'s comments (1/2)**
>
> **[Authors]** First of all, we really appreciate your careful comments. We address comments as below:
>
> ## Weaknesses:
> ---
> • This article argues that variable-rate compression methods like [R3.3] may increase the compression complexity by adding additional modules to the base model, so the author chooses to implement selective compression on the adaptive-quantization-based method. Also, the author chooses the model mentioned in [R3.1] and [R3.2] as the base model. However, the overhead complexity of the added module may not be that high compared to the base model itself. Also, it lacks a comparison between the proposed method and other variable-rate models which add modules to the base model in compress efficiency and complexity.
>
> **[Authors Reply 3-1]** Thanks for the comment. The complexity increase from additional modules may be significant or small depending on their architectures. Our point is that although our method uses additional network modules, the encoding/decoding time is reduced due to selective compression. In the experiments, we obtained 11.28% and 8.32% of decoding time reduction when Hyperprior and Mean-scale are used as base models, respectively. We additionally measured the encoding time reduction of 17.69% and 6.96% when Hyperprior and Mean-scale are used as base models, respectively, to reflect the comment by Reviewers 2.
>
> Regarding the comparison with the variable-rate models [19], the SCR models ($SCR_{Hyp}$ and $SCR_{MS}$ ) without selective compression are actually Cui et al.’s 2020 models [19], which are referred to as adaptive quantization in the manuscript. As shown in Fig. 6 and Fig. 10 in our manuscript, our SCR method is significantly superior to the Cui et al.’s 2020 models [19] in terms of coding efficiency for both MSE and MS-SSIM optimized models. In addition, when we visually compare the RD curves of our SCR models with those plotted in [19], all the SCR models ($SCR_{Hyp}^{psnr}$, $SCR_{Hyp}^{ssim}$, $SCR_{MS}^{psnr}$, $SCR_{MS}^{ssim}$, $SCR_{Cxt}^{psnr}$, and $SCR_{Cxt}^{ssim}$) significantly outperform the corresponding models in [19]. Please also refer to [Author Reply 1-6].
>
> Regarding the comparison with the variable-rate models [23], since the open source codes for their corresponding models are not available, we were not able to reproduce their RD curves. However, in comparison of the variable-rate models [23] (denoted as ‘SF in Balle (2018)’) with respective to the baseline model Balle (2018), it can be visually noticed in Fig. 10 (MS-SSIM) of our submitted supplemental that our $SCR_{Hyp}^{ssim}$ method based on Balle (2018) is significantly superior to the ‘SF in Balle (2018)’ in Fig. 6-(b) of the variable-rate model [23]. In the comparison of ‘SF in Balle (2018)’ in Fig. 6-(a) of the variable-rate model [23] against our $SCR_{Hyp}^{psnr}$ model in Fig. 6 of our main manuscript, it can be visually noticed that our SCR model outperforms the ‘SF in Balle (2018)’ in low (0.2bpp) and high (0.85 bpp) bit rate ranges. (Actually, 0.85 bpp is in the mid-range of our SCR models.)
>
> In addition to the coding efficiency comparisons, one thing to note is that our SCR method covers a much wider bit rate range compared to the other methods [19, 23]. The variable bit rate range supported by our method is approximately from 0.1 to 1.75bpp (PSNR 28dB ~ over 40dB), whereas the other two methods [19] and [23] support up to 1.0 and 0.85 bpp, respectively.  Please consider the supported bit rate range to be another key metric of the variable-rate compression performance.
>
> As the reviewer may also know, the learned image compression papers very often compare the performances of different methods by taking the values reported in the papers, not reproducing their proposed methods if the source codes are not available. Unfortunately, the numeric results of the variable-rate compression models [19, 23], which are based on the ref. compression models we also use, are not publicly open at the moment. We kindly requested the authors for the test results of the previous model [19] that the proposed SCR model is based on, but couldn’t receive a response yet. Thus we compared our SCR model with the reproduced version of [19] (SCR models w/o selective compression). Very recently, we also requested the authors of [23] for their test results. We’ll include more comparison results in the revised manuscript if possible.

---

> ### Author Response · Authors · 2022-08-09
> **Additional response to Reviewer 3 (mgum)'s comments**
>
> We received the experimental results of Chen [23] from the authors. On the Hyperprior [6] architecture, Our SCR method outperforms Chen [23] by 13.42% (1.64%) for the MS-SSIM (MSE)-optimized model in terms of MS-SSIM (PSNR) BD-rate. It should be noted that our SCR method not only outperforms Chen [23] in coding efficiency but also supports more than twice the bit-rate ranges compared to those in Chen [23], as below:
>
> <$SCR_{Hyp}$>
> - 0.209 to 1.819 bpp for the MS-SSIM-optimized model
> - 0.152 to 1.683 bpp for the MSE-optimized model
>
> <"SF in Balle (2018)" in Fig. 6 of Chen [23]>
> - 0.250 to 0.927 bpp for the MS-SSIM-optimized model
> - 0.187 to 0.837 bpp for the MSE-optimized model.
>
> For your information, in the comparison, we used the four target quality levels (q = 1.0, 3.0, 5.0, and 6.0) of our SCR models to match the bit-rate ranges as much as possible. When we used the whole range of our SCR models with q = 1.0, 4.0, 6.0, and 8.0, we obtained slightly greater BD-rate gains.
>
> We'll add these comparison results in the revised manuscript, and we would appreciate it if you could consider these comparison results as well.

---

### Official Review · Reviewer_DRRT · 2022-07-11

**Rating:** 5
**Confidence:** 4
**Soundness:** 2 fair
**Presentation:** 2 fair
**Contribution:** 2 fair

**Summary:**

- The authors propose NN-based variable-rate image compression method that selectively compresses the representations with generated 3D binary mask.

**Questions:**

- Is it possible to show the encoding time and its computing overhead when the proposed method is applied?
- Figure 6 shows that both SCR (w/o selective compression) and SCR (Full) above 0.6 bpp and below 1.2 bpp have slightly better R-D performance than baseline [6], and above 1.5 bpp, the R-D performance appears to be slightly worse than baseline [6]. Similar characteristics are exhibited for the baseline [7]. Why this characteristic?
- As a direct comparison with previous work on variable-rate compression, is it possible to experimentally compare the R-D curve with previous work (e.g. [19][23])?

**Limitations:**

- In practice, I believe there are many different neural image compression network architectures. I thought it would be good to discuss the architectures to which this method cannot be applied (e.g., without using hyperprior) and future works.

**Strengths And Weaknesses:**

Strengths
- Figure 6 shows that the proposed selective compression scheme appears to improve R-D performance at low rates compared to without it.
- According to Figure 9, the proposed method seems to have an advantage in decoding time by reducing entropy decoding time.
- Figure 3 provides an interesting visual representation of the importance of features.

Weaknesses
- The authors mention that the conventional method [1, 17, 18, 19, 20, 21, 22] requires additional network modules, layers, and inputs, which may result in complex overhead. But the proposed method may also requires additional processing, the authors present experimental results for processing time with respect to decoding, but not for encoding.
- There have been several studies in the past to generate importance maps for representations in deep image compression [a, b]. These are not cited in the current paper, but comparison with these would be good. Also, since the supp. material in [b] mentions the application of the importance map to multiple compression rates, I think it would be good to clarify the novel point and superiority of the proposal.
- It would be great to make direct experimental comparison of R-D curve or B-D rate with previous studies of variable-rate compression in deep image compression [1, 17, 18, 19, 23, 20, 21, 24, 22] with the authors' proposed method and demonstrate superiority.

[a] M. Li, et al. "Learning Convolutional Networks for Content-weighted Image Compression", CVPR 2018

[b] F. Mentzer, et al. "Conditional Probability Models for Deep Image Compression", CVPR 2018

---

> ### Author Response · Authors · 2022-08-02
> **Response to Reviewer 2 (DRRT)'s comments  (3/3)**
>
> ## Limitations:
> ---
> • In practice, I believe there are many different neural image compression network architectures. I thought it would be good to discuss the architectures to which this method cannot be applied (e.g., without using hyperprior) and future works.
>
> **[Authors Reply 2-7]** Thanks for the comment. We have already described the possible architectures that our method can be applied in line 105 (“compression architectures with hyper-encoder and hyper-decoder”), but we’ll also add some discussion on how our method can be incorporated into other architectures that do not use hyper-encoder and hyper-decoder. In the proposed SCR, we utilize the existing hyper en/decoder networks to compress/reconstruct (generate) the 3D importance maps. If some compression architectures do not include the hyper en/decoder, as an alternative, the en/decoder networks dedicated to the 3D importance maps can be used. In this case, we expect that the inherent compression efficiency of the variable bit rate model can almost be maintained due to selective compression because the bitstream for the auxiliary information is usually quite small. For instance, the bitstream from the hyperprior in the current SCR model only occupies 2~3% of the total bit rate, so the compressed side information only for the 3D importance map is expected to be less than this. However, the dedicated en/decoder networks may increase complexity overhead, so further study is needed on how to construct the dedicated en/decoder networks. To reflect the reviewer’s comment, we’ll discuss how to apply our SCR method for the compression architectures without hyper en/decoders in the revised manuscript.

---

> ### Author Response · Authors · 2022-08-02
> **Response to Reviewer 2 (DRRT)'s comments  (2/3)**
>
> • It would be great to make direct experimental comparison of R-D curve or B-D rate with previous studies of variable-rate compression in deep image compression [1, 17, 18, 19, 23, 20, 21, 24, 22] with the authors' proposed method and demonstrate superiority.
>
> **[Authors Reply 2-3]**  Thanks for the comment. As we mentioned in lines 235-238, most of the previous variable-rate compression methods [1, 17, 18, 20, 21, 22] are based on their own dedicated architectures which are not a widely used backbone architecture that our SCR method is based on. So, it was not able to make direct comparisons. Moreover, to our knowledge, there are not open source codes available for those prior works, which makes the reproduction of those models very hard. For these reasons, the studies on variable-rate compression usually conduct experiments with an emphasis on how well they reproduce the performance of multiple separately trained models using a single model. We kindly requested the authors for the test results of the previous model [19] that the proposed SCR model is based on, but couldn’t receive a response yet. Thus we compared our SCR model with the reproduced version of [19] (SCR models w/o selective compression). Very recently, we also requested the authors of [23] for their test results. We’ll include more comparison results in the revised manuscript if possible. Actually, we visually compared our experimental results (R-D Curves) with theirs. Please refer to [Authors Reply 3-1].
>
> In addition, the variable bit rate range supported by our SCR method is approximately from 0.1 to 1.75bpp (PSNR 28dB ~ over 40dB), which is significantly wider than the previous methods ([19]: ~1.0bpp, [23]: ~0.85bpp). Considering the purpose of the study, we think the supported bit rate range can be another key metric of the variable-rate compression performance.
>
> [a] M. Li, et al. "Learning Convolutional Networks for Content-weighted Image Compression", CVPR 2018
>
> [b] F. Mentzer, et al. "Conditional Probability Models for Deep Image Compression", CVPR 2018
>
> ## Questions:
> ---
> • Is it possible to show the encoding time and its computing overhead when the proposed method is applied?
>
> **[Authors Reply 2-4]** Thanks for the comment. Yes. Please refer to [Authors Reply 2-1].
>
> ---
> • Figure 6 shows that both SCR (w/o selective compression) and SCR (Full) above 0.6 bpp and below 1.2 bpp have slightly better R-D performance than baseline [6], and above 1.5 bpp, the R-D performance appears to be slightly worse than baseline [6]. Similar characteristics are exhibited for the baseline [7]. Why this characteristic?
>
> **[Authors Reply 2-5]** Thanks for the comment. Since our SCR method supports compression of a wide bit rate range using a shared single encoder/decoder network, it is even challenging to achieve the comparable performance against its separately trained models at their respectively dedicated target bit rates. Such a small inferiority in the high bit rate range is a result of joint optimization for multiple target quality levels, and the gain obtained from the rest compression points may be greater than the loss at the high bit rate from the total R-D loss (Eq. (4)) perspective.
>
> Regarding the mid-range compression efficiency of the reference model [6] where its performance is slightly worse than the proposed $SCR_{Hyp}^{psnr}$ model, one possible reason is an insufficient number of model parameters used for the mid-range baseline models. Also, the proposed selective compression may improve coding efficiency in that bit-rate range. Every representation component of the baseline model occupies a bit-rate more or less (although some of them occupy a very tiny amount), whereas the proposed SCR skips some unimportant representation components. This representation skipping mechanism of the SCR would lead to better coding efficiency in some bit-rate ranges.
>
> ---
> • As a direct comparison with previous work on variable-rate compression, is it possible to experimentally compare the R-D curve with previous work (e.g. [19][23])?
>
> **[Authors Reply 2-6]** Thanks for the comment. Yes. We’ll add comparison results when we receive the experimental results from the authors. Please refer to [Authors Reply 2-3].

---

> ### Author Response · Authors · 2022-08-02
> **Response to Reviewer 2 (DRRT)'s comments  (1/3)**
>
> **[Authors]** First of all, we really appreciate your careful comments. We address comments as below:
>
> ## Weaknesses
> ---
> • The authors mention that the conventional method [1, 17, 18, 19, 20, 21, 22] requires additional network modules, layers, and inputs, which may result in complex overhead. But the proposed method may also requires additional processing, the authors present experimental results for processing time with respect to decoding, but not for encoding.
>
> **[Authors Reply 2-1]** Thanks for the comment. The mentioned point is that although our method uses additional network modules, the decoding time is reduced. With respect to encoding time, it is also decreased due to selective compression as done for the decoder. Specifically, when using Hyperprior base model, the $SCR_{Hyp}^{psnr}$ (full) model reduces the encoding time by average 17.69% compared to the $SCR_{Hyp}^{psnr}$ (w/o SC) model, and average 16.56% compared to the Hyperprior reference model (corresponding to q=8.0). When Mean-scale model is used, $SCR_{MS}^{psnr}$ (full) model reduces encoding time by average 6.96% compared to $SCR_{MS}^{psnr}$ (w/o SC) model, and average 7.72% compared to Mean-scale ref. model (corresponding to q=8.0). We’ll add these comparison results for encoding time in the appendix of the revised manuscript.
>
> ---
> • There have been several studies in the past to generate importance maps for representations in deep image compression [a, b]. These are not cited in the current paper, but comparison with these would be good. Also, since the supp. material in [b] mentions the application of the importance map to multiple compression rates, I think it would be good to clarify the novel point and superiority of the proposal.
>
> **[Authors Reply 2-2]**  Thanks for the comment. It is important to note that the mentioned prior works with 2D importance maps are completely different with our 3D importance maps in perspectives of usage purpose and operations, which is not just a matter of 2D versus 3D importance maps. The purpose of using the 2D importance map in [a,b] is not to support variable-rate compression but to adptively alloate different bits for different regions according to their spatial importance. In this regard, they compress fewer (more) representation components for less (more) important spatial points. For a given bit budget, this is a matter of how to spread it over the spatial regions. However, in our problem, the 3D importance map represents the underlying component-wise importance of representations, which is a more generalized form of importance representation. For example, in [a,b],  when an importance value of a certain spatial point in the 2D importance map indicates “10 channels”, 10 adjacent representations in the channel axis (from channel index 0 to 9) at that spatial point are compressed. Whereas, our proposed SCR method selects the most beneficial components from an R-D perspective without such a constraint on channel orders.
>
> Although an extended application of the 2D importance map for multiple-rate compression was mentioned in the supplemental material of [b], we do not think it undermines the novelty of our study in that:
>
> i) the work of [b] has simply shown a possibility of multiple-rate compression by only showing the several decoded images without presenting any BD-rate performance for different degrees of the 2D importance maps; and,
>
> ii) in their preliminary experiments for multi-rate compression, their compression networks are trained to yield as many as the multiple 2D importance maps corresponding to the number of required target quality levels. Thus, the more target quality levels are, the more importance maps are needed to be envolved in the encoding process (because their multiple 2D importance maps are generated together within the encoder network).
>
> On the other hand, in our proposed SCR method, only one 3D importance map is used for all target quality levels because it represents the underlying inherent importance of each representation component. So our scheme can keep the simplicity of the variable-rate scheme even for the continuous bit-rate compression.
> We will revise the submitted manuscript to more clearly clarify our contribution with respect to [a, b], by differentiating our contribution in terms of the 3D importance map.

---

> ### Author Response · Authors · 2022-08-09
> **Additional response to Reviewer 2 (DRRT)'s comments**
>
> We received the experimental results of Chen [23] from the authors. On the Hyperprior [6] architecture, Our SCR method outperforms Chen [23] by 13.42% (1.64%) for the MS-SSIM (MSE)-optimized model in terms of MS-SSIM (PSNR) BD-rate. It should be noted that our SCR method not only outperforms Chen [23] in coding efficiency but also supports more than twice the bit-rate ranges compared to those in Chen [23], as below:
>
> <$SCR_{Hyp}$>
> - 0.209 to 1.819 bpp for the MS-SSIM-optimized model
> - 0.152 to 1.683 bpp for the MSE-optimized model
>
> <"SF in Balle (2018)" in Fig. 6 of Chen [23]>
> - 0.250 to 0.927 bpp for the MS-SSIM-optimized model
> - 0.187 to 0.837 bpp for the MSE-optimized model.
>
> For your information, in the comparison, we used the four target quality levels (q = 1.0, 3.0, 5.0, and 6.0) of our SCR models to match the bit-rate ranges as much as possible. When we used the whole range of our SCR models with q = 1.0, 4.0, 6.0, and 8.0, we obtained slightly greater BD-rate gains.
>
> We'll add these comparison results in the revised manuscript, and we would appreciate it if you could consider these comparison results as well.

---

> > ### Comment · Reviewer_DRRT · 2022-08-09
> > **Thank you for your explanation**
> >
> > Your explanation has helped me to dispel some concerns, such as encoding time and comparison with a few previous works ([19] [23]).
> > Additional explanations by the authors show that the proposed method seems to more generalized form than [a, b] and qualitatively more flexible. However, it is a bit unclear how much it makes a difference quantitatively in terms of RD-rate, etc. However, the perspective of faster entropy coding is not mentioned in [a, b], so in that respect, I think the methods and experimental results in this paper are valuable. But, the degree of speed-up may depend on the details of implementation.
> > And the comparison of RD-rate, BD-rate, speed and bit-rate range with the previous works would be great to be summarized and explained clearly in a table or graph.
> > Finally, assuming that the authors have revised the manuscript as per their response, I will increase the paper rating to "Borderline accept".

---

> > > ### Author Response · Authors · 2022-08-09
> > > **We appreciate your careful comments**
> > >
> > > We appreciate your careful comments and the adjustment of the rating.
> > >
> > > The previous methods [a, b] are mainly fixed-rate compression methods that for which direct comparisons are not applicable with our variable-rate compression (SCR) method. To make it worse, their compression architectures were of their own and are not based on common architectures (Hyperprior, Mean-scale, and Context), which makes unfair the comparison even in fixed rates.
> > >
> > > Even if the 2D importance map in [a, b] is applied for variable rate compression, it inclusively selects feature elements from low to high bit rates. That is, the selected features in a lower bit rate are all used in a higher bit rate. On the other hand, our 3D importance map represents the underlying component-wise importance of feature elements, independently of different target quality levels (different bit rates). Since this 3D importance map is adjusted channel-wise and then binarized into the target quality-dependent 3D binary mask, the selected features in a lower bit rate are not necessarily to be selected in a higher bit rate, which is more effective in the perspective of compression. This is observed in Fig. 5.
> > >
> > > As recommended, we will summarize and add a new table that tabularizes the BD-PSNR, BD-rate, encoding/decoding speeds, bit-rate ranges, etc. in the revised manuscript.

---

### Official Review · Reviewer_AG9W · 2022-07-11

**Rating:** 6
**Confidence:** 5
**Soundness:** 3 good
**Presentation:** 3 good
**Contribution:** 3 good

**Summary:**

This paper presents a method to improve the coding efficiency of the baseline variable bitrate model based on adaptive quantization (quantization steps controlled by quality scaling factors) while at the same time reducing entropy coding time, thus one step closer to closing the gap to the fixed bitrate models. Specifically, it proposes to generate 3D importance binary map for the latents to be selectively coded. The 3d map generation is based on side information (decoded hyper latent) and a target bitrate. Element-wise mask without much overhead is the key to bridge the gap to fixed bitrate models.

**Questions:**

1. please confirm that even through results in Cui 2020 (e.g. Fig 2) looks like it already closes the gap to fixed rate reference models, according to your evaluation, it is still worse at lower bitrates. As stated in line 87-88?
2.  For your ablation, what is exactly `the SCR variant without the selective compression`?
3.  Do you have ablation on the importance adjustment? e.g. with only adaptive quantization from Cui 2020 + 3d importance map; or quality q as input to 3d map generation besides the second to last features in hyper decoder.
4.  line 126: just to clarify, for mean-scale hyperprior, masked out latent is just set to mean value predicted by hyper decoder, right? same for line 253.
5.  Do you also have decoding time profiling for Context-based models? it is not shown in Sec 4.2
6. Fig 9:
   * the Y-axis is second?
   * why reshape time increases as rate goes up?
   * why hyper decoder takes comparable/ much longer than decoder network? It should be around/less than 1/10 of the decoder?
   * 15ms for the decoder (yellow) seems too small for Kodak images. What GPU exactly are you profiling on?
   * any details on entropy coding? any parallel impl?




**Limitations:**

- The paper does not mention the *memory* increase due to generation of 3D importance map in encoding / decoding. Should profile this in decoding time section as well.
- I like the study in figure 5. Is it possible to extend to embedded bitstream? i.e. encode an image to a single bitstream, to get lower bitrates, just truncate to a prefix of this stream.

**Strengths And Weaknesses:**

Strengths
- Almost close the gap between variable bitrate models with separately optimized fixed-bitrate models with element-wise 3D importance map. The RD improvement over the baseline method (Cui 2020) mainly over the lower bitrates is convincing.
- The techniques induces small overhead (a single 1x1 conv, and parameterized importance adjustment curves), can be applied to many existing models, as showed in the paper: hyperprior, mean-scale, context-based.
- The paper is well organized with detailed description of various parts of the whole compression system.

Weaknesses
- Lack of ablation on the importance adjustment, which is one of the main contribution of this paper.
- The presentation on reducing decoding time in Sec 4.2 is a bit misleading. line 295 - "Furthermore, SCRpsnr Hyp294 and SCRpsnr MS showed lower decoding times compared to the original reference compression models over all target quality levels except q = 8..." Fig  9 only shows the decoding time for reference models for q=8 instead of all target quality levels.
- Also the way to show comparison results with BD rate can be more fair when all variable bitrate models are compared to the reference model, instead of like line 248-249.
- The third contribution claim on continuous bitrate through interpolation already exists in Cui 2020?
- Training procedure becomes more complicated and expensive - Three stages, in total 9.4 M steps.

---

> ### Author Response · Authors · 2022-08-02
> **Response to Reviewer 1 (AG9W)'s comments (3/3)**
>
> o why hyper decoder takes comparable/ much longer than decoder network? It should be around/less than 1/10 of the decoder?
>
> **[Authors Reply 1-13]** Thanks for the detailed comment. This is because we use a CPU (not GPU) for hyper-decoder processing. If we use GPU for the hyper decoder processing, the encoder and decoder discrepancy occurs owing to the GPU characteristic that incurs tiny errors for the same inputs even on the same machine, because of its parallel algorithm that yields different numeric results [A]. On the other hand, the decoder network processing is free from such an encoder/decoder discrepancy, so we used a GPU. If we had been able to process the hyper decoder with the GPU, the time complexity gain would have been significantly increased.
>
> ---
> o 15ms for the decoder (yellow) seems too small for Kodak images. What GPU exactly are you profiling on?
>
> **[Authors Reply 1-14]** Thanks for the detailed comment. We think that 15ms for the decoder network is not specifically small but comparable to the results reported in other articles. As shown in Fig.1 of He et al. [B], the total decoding time of the Hyperprior(indicated as “Balle2018”) model is around 17 ms in their experiments. For our experiments, we used 1 x RTX Titan GPU.
>
> ---
> o any details on entropy coding? any parallel impl?
>
> **[Authors Reply 1-15]** Thanks for the detailed comment. For Hyperprior, Mean-scale, and their corresponding variable-rate models, we used the entropy coding module in “tensorflow-compression v1.3” [C]. Although we didn’t implement the parallel processing by ourselves, the tensorflow-compression package [C] partially supports the parallel processing for entropy coding and decoding, to our knowledge. For the Context and their variable-rate models, we adopted the python-based entropy coder [D] to measure bpp values based on stored bitstream files, rather than the string list level. When using the entropy coder in the tensorflow-compression package [C] for the Context model, we can obtain the entropy coded bitstream (string) per each spatial point of the latent representation. With these point-wise strings, we have to measure the bpp values based on the total summation of the string sizes. However, when this string list is stored as a file, the saved file size must be larger than the sum of the string sizes, because each string per position has a different string length. To overcome this implementation issue, we adopted the entropy coder [D] for the Context model, although the entropy coder [D] is very slow without any parallel processing.
>
> ---
> [A] https://docs.nvidia.com/cuda/floating-point/index.html
>
> [B] He, et al. “ELIC: Efficient Learned Image Compression with Unevenly Grouped Space-Channel Contextual Adaptive Coding”, CVPR2022
>
> [C] https://github.com/tensorflow/compression/tree/v1.3
>
> [D] https://github.com/JooyoungLeeETRI/CA_Entropy_Model
>
> ## Limitations:
> ---
> • The paper does not mention the memory increase due to generation of 3D importance map in encoding / decoding. Should profile this in decoding time section as well.
>
> **[Authors Reply 1-16]** Thanks for the detailed comment. The increase in model parameters due to the proposed method is very limited. For example, the Hyperprior model has 13,042,371 parameters, and the $SCR_{Hyp}$ (full) model has 13,111,811. That is, the number of parameters is increased by only 0.53%, resulting in a memory increase of 0.26MB during en/decoding. Specifically, the configurations of the increased parameters are shown below:
>
> ||description|#param|
> |---|---|---|
> |Hyperprior|original param. Size|13,042,371|
> |QV vectors|8 levels * 320 ch|2,560|
> |IQV vectors|8 levels * 320 ch|2,560|
> |gamma vectors|8 levels * 320 ch|2,560|
> |importance map gen. (1x1 conv.)|192 (in) * 320 (out)|61,440|
> |1x1 conv. Bias|320 (out)|320|
>
> Similarly, the additional memory occupancy required for the Mean-scale and Context model is only 0.62MB. We will describe the increases in memory sizes and parameter numbers in the revised manuscript.
>
> ---
> • I like the study in figure 5. Is it possible to extend to embedded bitstream? i.e. encode an image to a single bitstream, to get lower bitrates, just truncate to a prefix of this stream.
>
> **[Authors Reply 1-17]** Thanks for the comment. As shown in Fig. 5, in the proposed method, most of the selected latent representations are also utilized at a higher target quality level, so we expect that the proposed method can be extended to the quality-scalable image compression codecs. However, since the proposed model is a combination of selective compression and adaptive quantization, it is necessary to address quality-specific quantization issues along with the truncation of the representation components.

---

> ### Author Response · Authors · 2022-08-02
> **Response to Reviewer 1 (AG9W)'s comments (2/3)**
>
> 2. For your ablation, what is exactly the SCR variant without the selective compression?
>
> **[Authors Reply 1-7]** Thanks for the comment. As described in line 234, “the SCR variant without the selective compression” is the variable-rate compression model only using the adaptive quantization based on QV_q and IQV_q vectors.
>
> ---
> 3. Do you have ablation on the importance adjustment? e.g. with only adaptive quantization from Cui 2020 + 3d importance map; or quality q as input to 3d map generation besides the second to last features in hyper decoder.
>
> **[Authors Reply 1-8]** Thanks for the comment and some ideas. As replied in [Authors Reply 1-1] response, the proposed importance adjustment method is very simple and works reasonably well in our study. It is worthwhile to further improve the importance adjustment method with various trials, and we’ll carefully analyze them in future works.
>
> Regarding the idea that feeds the target quality levels together into the importance map generation module, we think it would be one good candidate and can be a more generalized form of importance adjustment. However, if we maintain the current simple architecture of the 3D importance map generation module, which is a single 1x1 conv. layer, the target quality level will be involved in only one weighted summation, and that may not be sufficient. To deal with the input of target quality levels, we may need a deeper network or a different architecture based on vector embedding of the target quality level followed by (linear or non-linear) scaling of the generated importance. As mentioned above, we’ll further analyze these in the future. Thanks.
>
> ---
>
> 4. line 126: just to clarify, for mean-scale hyperprior, masked out latent is just set to mean value predicted by hyper decoder, right? same for line 253.
>
> **[Authors Reply 1-9]** Thanks for the detailed comment. For mean-scale hyperprior, the unselected representations are also filled with 0 values. We also tested the version that fills the unselected representations with estimated mean values, but the results were almost identical. For the model simplicity, we use 0 values instead of the estimated mean values. We’ll more clearly describe this in the revised manuscript.
>
> Regarding the comment on the line 253, the unselected representations were also filled with 0 values for the $SCR_{Cxt}$ models, because of a circular dependency issue (chicken-and-egg problem) in training. More specifically, this circular dependency issue is due to the masked convolutions used for the mean and variance estimation in the training phase. To “properly” estimate the mean with the masked convolution, the unselected representation components must be already filled with the “properly” estimated mean values. That is, the mean estimation and the preparation of context representations are mutually dependent. From our understanding, the only way to avoid this circular dependency is to build the whole autoregression chain for training instead of using the masked convolution, but it’s almost impossible because a huge amount of memory is required.
>
> ---
> 5. Do you also have decoding time profiling for Context-based models? it is not shown in Sec 4.2
>
> **[Authors Reply 1-10]** Thanks for the detailed comment. Yes, we do have. For Context-based models, the $SCR_{Cxt}^{psnr}$ (full) model reduces the decoding time by 25.71% compared with the $SCR_{Cxt}^{psnr}$ (w/o SC) model, and by 21.53% compared with the ref. Context model (corresponding to q=8.0). In the case of encoding, the $SCR_{Cxt}^{psnr}$ (full) model reduces the encoding time by 20.25% compared with the $SCR_{Cxt}^{psnr}$ (w/o SC) model, and by 13.90% compared with the ref. Context model (corresponding to q=8.0).
>
> Despite the greater reduction ratio in encoding and decoding time, we didn’t include these results in the first manuscript we submitted because we had a small concern that time measurements using the entropy coder different from those used in the Hyperprior-based and Mean-scale-based models might cause confusion. However, since these results also show the effectiveness of selective compression, we’ll add them with detailed descriptions in the revised manuscript. Please also refer to [Authors Reply 1-15].
>
> ---
> 6. Fig 9:
>
> o the Y-axis is second?
>
> **[Authors Reply 1-11]** Thanks for the detailed comment. Yes. it’s second. We’ll add it in the revised manuscript.
>
> ---
> o why reshape time increases as rate goes up?
>
> **[Authors Reply 1-12]** Thanks for the detailed comment. That’s because the proportion of selected representations (input) increases as the rate increases. As shown in line 434 in Appendix A, the proposed model needs to allocate more representation components in place for higher target quality levels.

---

> ### Author Response · Authors · 2022-08-02
> **Response to Reviewer 1 (AG9W)'s comments (1/3)**
>
> **[Authors]** First of all, we really appreciate your careful comments. We address comments as below:
>
> ## Weaknesses
> ---
> • Lack of ablation on the importance adjustment, which is one of the main contribution of this paper.
>
> **[Authors Reply 1-1]** Thanks for the comment. Please note that our importance adjustment is used to adaptively generate the 3D binary mask according to given target quality levels. To do so, we utilize the channel-wise gamma correction curves that are widely used for adjusting the intensity of images and videos, based on the assumption that the representation components in the same channel follow a similar distribution. According to the given target quality level, the corresponding importance adjustment is performed on the underlying 3D important map that is independent of the target quality levels. In this regard, the importance adjustment is an essential component to support the different target quality levels, which is not the component to be ablated to see the resulting performance without itself. If the reviewer’s comment means the lack of other trials for the importance adjustment, we actually tried several methods and our proposed importance adjustment method turned out to perform the most reasonably well, considering that the importance adjustment curves can adequately amplify or attenuate the 3D importance map values according to the given target quality levels, by simply using 8 importance adjustment vectors.
>
> ---
> • The presentation on reducing decoding time in Sec 4.2 is a bit misleading. line 295 - "Furthermore, SCRpsnr Hyp294 and SCRpsnr MS showed lower decoding times compared to the original reference compression models over all target quality levels except q = 8..." Fig 9 only shows the decoding time for reference models for q=8 instead of all target quality levels.
>
> **[Authors Reply 1-2]** Thanks for the detailed comment. We will clearly specify that our method is compared with the reference model corresponding to q=8.0.
>
> ---
> • Also the way to show comparison results with BD rate can be more fair when all variable bitrate models are compared to the reference model, instead of like line 248-249.
>
> **[Authors Reply 1-3]** Thanks for the comment. We actually measured and presented the BD-rate values for four target quality levels with q = 1.0, 4.0, 6.0 and 8.0. Please see the lines 244-266.  For more clarification, we will revise the sentence by indicating that the presented BD-rate values were measured with the target quality levels of q = 1.0, 4.0, 6.0 and 8.0 of our SCR models and the corresponding four compression points of the reference models. Please note that the BD-rate is not measured one-by-one but is calculated as a whole for the four target quality levels.
>
> ---
> • The third contribution claim on continuous bitrate through interpolation already exists in Cui 2020?
>
> **[Authors Reply 1-4]** Thanks for the comment. Cui 2020 introduced the interpolation of parameters that determine quantization and inverse quantization steps to support the continuous bitrates. Our SCR extends Cui’ interpolation-based approach by additionally incorporating the interpolation of the importance adjustment curves. In experiments, we verified that the interpolation can stably support the continuous bitrate compression. We will clarify this point in our contribution.
>
> ---
> • Training procedure becomes more complicated and expensive - Three stages, in total 9.4 M steps.
>
> **[Authors Reply 1-5]** Thanks for the comment. The training of variable rate compression models often requires much longer training time than single-quality models. To alleviate this, we adopted a step-wise training to reduce the training time, which is somewhat complicated than the one-step training, but “less” expensive. Please note that the step-wise training is typical in the training of variable-rate compression models, and further study is required in the research domain for simple and low-cost training.
>
> ## Questions:
>
> ---
> 1. please confirm that even through results in Cui 2020 (e.g. Fig 2) looks like it already closes the gap to fixed rate reference models, according to your evaluation, it is still worse at lower bitrates. As stated in line 87-88?
>
> **[Authors Reply 1-6]** Thanks for the comment. The experimental of the reference models in Cui 2020 [19] were obtained using their reproduced versions by Cui et al. However, the reproduced performances for the reference models are significantly lower than those reported in the original papers of the reference models. From this, the performance of Cui’s model looks comparable to those of the reference models. In many papers, the performance results of the reference models are taken from their original papers without reproductions because the source codes may not be available and the exact implementations of the reference models are very difficult.

---

### Meta-Review · Area_Chair_ukBr · 2022-08-26

**Recommendation:** Accept
**Confidence:** Certain

**Metareview:**

Thanks for your submission to NeurIPS.

Initially, this paper was leaning reject, with three negative reviewers who had various concerns.  The rebuttal really helped a lot, and two of the negative reviewers raised their scores based on the rebuttal, leading to 3 of 4 accept scores.  The final reviewer also mentioned in a comment that they would raise their score to a borderline accept, but never updated the official score.  I also took a look at the reviews and rebuttal, and it seems that the major concerns have indeed been addressed.

Given all of this, I am happy to recommend acceptance of the paper at this point.

**Award:**

No

---

### Decision · Program_Chairs · 2022-09-14

Accept